# Nonparametric Unsupervised Data Condensation for Gigapixel Histological Images

## Abstract

Histological whole-slide images (WSIs) are central to computational pathology but are extremely large, often several gigabytes, making them infeasible for direct use in standard vision pipelines. Prior approaches reduce training cost by condensing WSIs into a fixed number of representative features (prototypes), but this approach overlooks the varying complexity and diversity of WSIs, leading to loss of critical information. To this end, we propose **NICER**, a probabilistic data condensation framework that decomposes each WSI into feature patterns to capture heterogeneity and concept prototypes to ensure compactness. By reformulating prototype construction as a nonparametric condensation problem, NICER adapts the number of prototypes to slide complexity while preserving relevant information. Experiments on four histological datasets show that NICER outperforms prior methods, yielding up to 90% performance gains and superior efficiency trade-offs, setting a new paradigm for histological representation learning.

## 1 Introduction

Histological whole-slide images (WSIs) are high-resolution digital scans of tissue slides and have become central to Computational Pathology (CPath) (Song et al., 2024; 2023), enabling tasks such as classification (Xiang & Zhang, 2023; Shao et al., 2021), segmentation (Graham et al., 2023; Guo et al., 2023), and survival prediction (Fan et al., 2023). However, their enormous resolution, often exceeding $100,000 \times 100,000$ pixels (hundreds of gigapixels and several gigabytes per slide), makes full-slide processing infeasible. For example, even a single WSI cannot fit into a multi-head self-attention (MHSA) unit due to its quadratic memory complexity.

**Challenge.** Multiple Instance Learning (MIL) addresses WSI scale by partitioning each slide into thousands of patches (e.g., $> 10,000$), embedding them with a pre-trained encoder, and aggregating them into a slide-level representation (Tang et al., 2023; Nguyen et al., 2025b; Xiang & Zhang, 2023). Since the full processing and storing of patches is costly (Jin et al., 2025; Sacco et al., 2020), recent studies reveal the strong morphological redundancy of WSIs (Song et al., 2024; Vu et al., 2023), summarize them into compact, representative prototype sets that are transferable for downstream tasks (Song et al., 2024; Jin et al., 2025). However, the high variability of WSIs undermines the representativeness of the prototypes. For instance, some slides contain large homogeneous regions, while others show highly heterogeneous tumor areas requiring denser sampling (see Fig. 1).

**Limitation of Prior Work.** Existing methods (Vu et al., 2023; Claudio Quiros et al., 2024; Song et al., 2024) operate under the restrictive assumption that a fixed set of prototypes can adequately represent all slides, regardless of their complexity. While conceptually simple, this assumption overlooks the wide variability in morphological redundancy and structural complexity across slides and institutions, leading to either redundant prototypes or information loss (see Fig. 1). Failing to adapt to this variability forces a trade-off between accuracy and efficiency, with most approaches sacrificing the former for the latter, as shown in Fig. 2.

**Fundamental Gap.** In hindsight, what is missing from existing approaches is a mechanism to balance aggressive feature reduction with information preservation in an unsupervised manner. Because feature distributions and complexity levels vary widely across slides, using a prototype set with fixed capacity may achieve efficiency but risks losing critical information or introducing redundancy. Increasing the prototype set size can mitigate information loss but at the cost of reduced efficiency for the entire WSI pipeline. This raises a fundamental question: *How can we identify and*

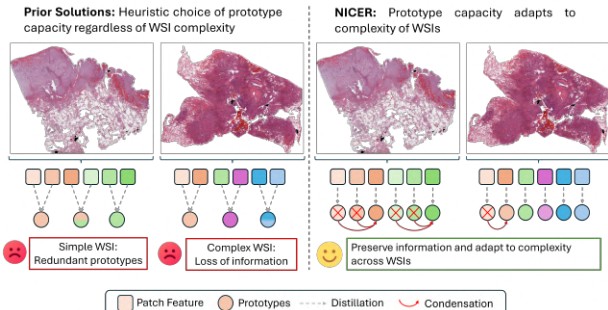

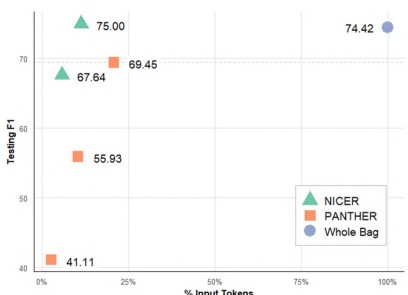

Figure 1: Conceptual illustration highlighting the key distinction between our NICER and prior work. When WSI's complexity exceeds the prototypes' capacity, different features (different colors) might collapse into less information prototypes, and simple WSIs might suffer from redundancy.

Figure 2: On the same task and model, NICER achieves higher compression rate and better downstream performance (F1) than other baselines. (see Sec. 3).

*model the varying complexity of WSIs during condensation?* Tackling this challenge hence calls for a new approach that adaptively balances information preservation and efficiency on a per-slide basis, enabling flexible and optimal slide-level representations.

**Solution Vision.** To address the above question, our solution insight is: effective WSI condensation should not begin with aggressive reduction, but rather with intentional redundancy to preserve rare and heterogeneous signals which vary significantly across WSIs. Redundancy is then adaptively removed to restore efficiency, allowing each slide to determine its own capacity based on complexity. This design avoids early information loss, achieves a principled balance between preservation and efficiency, and provides a flexible foundation for robust, slide-adaptive learning.

**Technical Contribution.** To realize this vision, we introduce **NICER**, a novel *NonparametrIC unsupERvised data condensation framework* that reformulates prototype construction as an unsupervised condensation problem. NICER first learns a high-capacity set of *feature patterns* to preserve diverse and heterogeneous information from each slide redundantly, and then condense them into a compact set of *feature concepts*. Redundant concepts are pruned, and the number of retained concepts adapts automatically to slide complexity, making the process nonparametric and slide-adaptive. The entire procedure is formalized through a generative formulation governed by learnable parameters. Our main contributions are as follows:

**1.** We cast prototype construction as an unsupervised data condensation task, formulated as a hierarchical optimization problem. Prototypical information is first distilled from the WSI feature bag into a set of patterns, which are then condensed into a compact set of concepts. This design adapts the concept set capacity to the complexity of each WSI, achieving a balance between information preservation and efficiency (see Section 2.3).

**2.** We develop an algorithm that identifies the most probable associations between patterns and condensed concepts. Framed as a latent variable in our generative model, this association is efficiently inferred in a probabilistic view, enabling the proposed approach to be practical and applicable across diverse real-world medical settings (see Section 2.4).

**3.** We evaluate the performance of NICER against existing baselines through extensive experiments on cancer subtyping and survival prediction tasks, spanning four benchmark datasets. The results demonstrate that NICER consistently surpasses competing methods across diverse settings, establishing new state-of-the-art performance in unsupervised prototype construction (see Section 3).

## 2 METHODOLOGY

### 2.1 PROBLEM FORMULATION AND METHOD OVERVIEW

Unlike prior work (Vu et al., 2023; Song et al., 2024), which prioritizes efficiency over information preservation by fixing prototype capacity across all WSIs, our goal is to balance the two in an un-

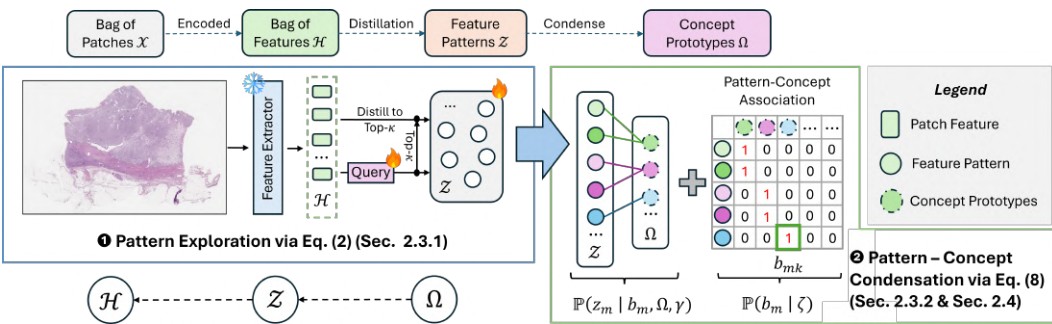

Figure 3: **Overview of NICER.** NICER decouples preservation and efficiency into patterns $\mathcal{Z}$ and concepts $\Omega$, learned via two stages: *pattern exploration* to capture diverse information, and *condensation* to merge redundancies into compact concepts.

supervised and slide-adaptive manner. To this end, we introduce a hierarchical formulation wherein each WSI is represented by a pattern set preserving the diversity of the original feature bag, which is subsequently summarized into more compact concepts as follows:

**Problem Formulation.** Formally, a WSI is tessellated into non-overlapping patches $\mathcal{X} = \{x_1, \ldots, x_N\}$, where each patch $x_i \in \mathbb{R}^{H \times W \times 3}$. A pretrained encoder $f_{\text{enc}}(\cdot)$ maps each patch $x_i$ to a latent embedding $h_i \in \mathbb{R}^d$, yielding a feature bag $\mathcal{H} = \{h_1, \ldots, h_N\}$. The objective is to condense $\mathcal{H}$ into a smaller concept (prototype) set $\Omega = \{\omega_k\}_{k=1}^K$ with $K \ll N$, i.e., maximizing $\mathbb{P}(\mathcal{H} \mid \Omega)$ under Bayesian lens. We rewrite this problem under the hierarchical abstraction $\mathcal{H} \leftarrow \mathcal{Z} \leftarrow \Omega$, where patterns $\mathcal{Z}$ approximate features $\mathcal{H}$ and concepts $\Omega$ are their underlying generators, leading to maximization of $\mathbb{P}(\mathcal{H}, \mathcal{Z} \mid \Omega)$.

This two-level abstraction provides a principled framework for efficient WSI condensation and explicitly balances accuracy and efficiency, addressing a key limitation in previous researches.

**Method Overview.** Fig. 3 illustrates the NICER framework, which operates in two iterative stages. In the *pattern exploration*, patterns $\mathcal{Z}$ are learned to capture the diversity of $\mathcal{H}$ through selective interactions between patches and patches (Sec. 2.3). In the *condensation stage*, concepts $\Omega$ are introduced to enforce compactness by modeling patterns as samples generated from a smaller concept set (Sec. 2.3.2). During condensation, concepts that do not contribute to generating observed patterns are pruned, imparting NICER with a nonparametric nature and scalability to the complexity of patterns. In summary, patterns ensure information preservation, while concepts enforce efficiency.

## 2.2 PROBABILISTIC NONPARAMETRIC DATA CONDENSATION

Intuitively, using proposed hierarchical formulation, NICER enriches the standard WSI condensation framework by inserting an intermediate random variable that helps preserve the diversity of features in the original WSI. Instead of collapsing the whole slide image into a single compressed set directly, we first allow each WSI to generate a pattern set of size $M$. These patterns ($\mathcal{Z}$) retain the originality of the data ($\mathcal{H}$) while naturally containing some redundancy. The condensation phase then acts like a sculptor, carefully shaving away overlaps and compressing the patterns into a compact, information-rich concept set $\Omega$.

**Data Condensation Model.** Given a hierarchical probabilistic model $\mathcal{H} \leftarrow \mathcal{Z} \leftarrow \Omega$, the data condensation model is factorized by:

$$\log \mathbb{P}(\mathcal{H}, \mathcal{Z} \mid \Omega) = \log \mathbb{P}(\mathcal{H} \mid \mathcal{Z}, \Omega) + \log \mathbb{P}(\mathcal{Z} \mid \Omega). \qquad (1)$$

Data condensation is then achieved via fitting the parameterization of this probabilistic model to the pre-trained feature observation $\mathcal{H}$, followed by the most probable concept set $\Omega = \arg\max_{\Omega'} \max_{\mathcal{Z}} \mathbb{P}(\mathcal{H}, \mathcal{Z} \mid \Omega')$ via principled probabilistic inference.

## 2.3 PARAMETERIZATION

We parameterize the conditional terms in the condensation model (Eq. 1), interpreting $\mathbb{P}(\mathcal{H} \mid \mathcal{Z}, \Omega)$ as *pattern exploration*, where diverse patterns ($\mathcal{Z}$, of size $M$) are extracted, and $\mathbb{P}(\mathcal{Z} \mid \Omega)$ as *condensation*, modeling patterns with an unknown concept set $\Omega$. This formulation enables nonparametric condensation and slide-wise adaptability.

### 2.3.1 PATTERN EXPLORATION

In the pattern exploration stage, we use $\mathbb{N}(h_i \mid \mu_i, \mathbf{I})$ to denote the density of $h_i$, which is modeled as a Gaussian variable $h_i \sim \mathbb{N}(\mu_i, \mathbf{I})$, with the mean $\mu_i \triangleq \mu_i(\mathcal{Z}; \theta)$ maps the shared pattern set $\mathcal{Z}$ into a representation best aligned with $h_i$. Conceptually, this is akin to a retrieval process: each feature selects its best-matching pattern, $\mu_i(\mathcal{Z}; \theta) = z^*_{(i)} = \mathcal{Z}[m^*_i]$ where $\mathcal{Z}[m^*_i]$ denotes the selected pattern for $h_i$ from $\mathcal{Z}$. When $h_i$ and $z_m$ are $\ell_2$-normalized, the log-likelihood reduces to

$$\log \mathbb{P}(\mathcal{H} \mid \mathcal{Z}, \theta) = \sum_{i=1}^{N} \log \mathbb{N}(h_i \mid z^*_{(i)}, \sigma^2 \mathbf{I}) \approx -\frac{1}{2\sigma^2} \sum_{i=1}^{N} \left( 2 - 2\langle h_i, z^*_{(i)} \rangle \right). \quad (2)$$

**Remark.** Eq. 2 reveals an intuitive principle: maximizing the likelihood of the feature bag $\mathcal{H}$ reduces to aligning features with their closest patterns. Thus, pattern learning naturally emerges as a retrieval-style process, but one firmly anchored in a probabilistic framework.

**Design and Learning.** Based on the above probabilistic analysis, we distill prototypical information from the feature bag $\mathcal{H}$ into the pattern set $\mathcal{Z}$ by framing distillation as a retrieval-based selection process. Each feature $h_i \in \mathcal{H}$ associates with pattern $z_m \in \mathcal{Z}$ via cosine similarity,

$$\Gamma(h_i, z_m) \triangleq \langle h_i, z_m \rangle. \quad (3)$$

Extending the baseline formulation in Eq. 2, we allow each feature to distribute its information across its top-$\kappa$ most relevant patterns ($\kappa \ll M$), which improves expressiveness while preventing over-dispersion that weakens discriminative power. Maximizing $\Gamma(\cdot, \cdot)$ then drives an adaptive selection process, yielding a pattern set $\mathcal{Z}$ that is both compressed and information-preserving, diverse enough to capture the variability of $\mathcal{H}$, and efficient for the subsequent condensation stage.

### 2.3.2 PATTERN-CONCEPT CONDENSATION

In this section, we derive the condensation term in Eq. 1 and present the construction of the concept set $\Omega$. While the pattern set $\mathcal{Z}$ provides a comprehensive view of $\mathcal{H}$, it remains tied to local feature variations and often carries redundancy. To move beyond this, we introduce $\Omega$ as a higher-level *semantic abstraction*, a compact set of concept prototypes that captures only the essential structures of the WSI, discarding spurious noise and redundant information. This step is formalized as a generative process, where each pattern in $\mathcal{Z}$ is modeled as a probabilistic sample from a concept prototype in $\Omega$, and the pattern-concept associations are inferred by maximizing the posterior distribution (Sec. 2.4 for details).

**Pattern Prior.** In particular, we model this condensation process as a generative model based on a nonparametric point process. In this view, the point process distributed a prior set of latent concepts, and each WSI pattern is considered as a sample from a pattern-generation distribution parameterized with a particular concept. We hence enforce that a pattern $z_m \in \mathcal{Z}$ must be generated by a distribution governed by exactly one concept prototype $\omega_k \in \Omega$ with $\omega_k$ indicates the $k$-th concept. Concretely, each $z_m \in \mathcal{Z}$ is treated as an independent sample from a Gaussian distribution:

$$z_m \sim \mathbb{N}\left( \psi_m(\Omega), \Sigma_m(\Omega, \gamma) \right) \quad (4)$$

where $\psi_m$ is an assignment neural function that determines if the concept prototype $\omega_k$ responsible for generating $z_m$ and $\Sigma_m(\cdot, \gamma)$ is neural function estimating corresponding covariance matrix with parameters $\gamma$. Since this generative model aims to perform condensation, each pattern $z_m$ must be assigned to exactly one concept prototype $\omega_k \in \Omega$. This prior is implemented explicitly on the assignment function $\psi_m$ via an introduction of novel assignment variables $b_m \triangleq (b_{m1}, b_{m2}, \ldots, b_{mK})$ such as $b_{mk} \in \{0, 1\}, \forall m = 1 \ldots M, k \ldots K$ and $\sum_k b_{mk} = 1$. Further diagonalizing the parameterized covariance matrix, we can rewrite Eq. 4 as follows:

$$z_m \mid b_m \sim \mathbb{N}\left( \psi_m, \mathrm{diag}\left( \delta\left( \psi_m; \gamma \right) \right) \right), \text{ where: } \psi_m \triangleq b_{m1} \cdot \omega_1 + b_{m2} \cdot \omega_2 + \ldots + b_{mK} \cdot \omega_K. \quad (5)$$

where $\Sigma_m(\Omega, \gamma)$ is rewritten to $\text{diag}(\delta(\Omega, \gamma))$ with $\delta(\Omega; \gamma)$ as a neural function with parameter $\gamma$. Here, $b_{mk}$ indicates whether $m$-th pattern is generated by $k$-th concept. This reformulation shows that each pattern associate with one concept in the learnable set $\Omega$ and their assignment is governed by binary variables $b_{mk}, \forall m \in \{1, \ldots, M\}$ and $\forall k \in \{1, \ldots, K\}$.

**Assignment Prior.** Using Eq. 4, Eq. 5 and the definition of our new assignment variable $b \triangleq \{b_m\}_{m=1}^M$, we can now solve Eq. 1 with respect to $\mathbb{P}(\mathcal{Z}, b \mid \Omega)$ instead of $\mathbb{P}(\mathcal{Z} \mid \Omega)$ as follows,

$$\log \mathbb{P}(\mathcal{Z}, b \mid \Omega) = \log \left\{ \prod_{m=1}^M \mathbb{P}(z_m, b_m \mid \Omega, \gamma, \zeta) \right\} = \sum_{m=1}^M \log \mathbb{P}(z_m \mid \Omega, \gamma) + \log \mathbb{P}(b_m \mid \zeta), \quad (6)$$

where $\mathbb{P}(b_m \mid \zeta)$ imposes an assignment prior governed by parameters $\zeta$. To ensure that every pattern $z_m \in \mathcal{Z}$ is consistently tied with exactly one concept $\omega_k \in \Omega$, we enforce a categorical distribution over the assignment variables $b_m$ as,

$$\mathbb{P}(b_m | \zeta) \triangleq \prod_{k=1}^K \pi_k^{b_{mk}} \text{ , where: } \pi_k \triangleq \frac{\exp(\alpha(\omega_k; \zeta))}{\sum_k \exp(\alpha(\omega_k; \zeta))} \quad (7)$$

where $\alpha(\cdot)$ is a deep network parameterized by $\zeta$. This prior serves two complementary purposes. First, the categorical form enforces that each pattern $z_m \in \mathcal{Z}$ is associated with exactly one concept prototype, thereby encouraging compactness in the condensation process. Second, by parameterizing the assignment probabilities through learnable logits, NICER adapts the allocation of patterns to concepts dynamically, ensuring that assignments reflect the diverse information captured in $\mathcal{Z}$.

## 2.4 CONDENSATION LEARNING

Given the nonparametric pattern-generating story above, our original objective which maximizes the joint likelihood of $\mathcal{H}, \mathcal{Z}$ given $\Omega$ (see Eq. 1) now reduces to the pattern condensation problem as,

$$\max_{\Omega, \gamma, \zeta, b} \left\{ \sum_{m=1}^M \log \mathbb{P}(z_m, b_m \mid \gamma, \zeta, \Omega) \right\} = \max_{\Omega, \gamma, \zeta, b} \sum_{m=1}^M \left\{ \log \mathbb{P}(z_m \mid b_m, \gamma, \Omega) + \log \mathbb{P}(b_m \mid \zeta) \right\} \quad (8)$$

which is directly computable when using Eq. 5 and Eq. 7 (see Appendix E for more details). Solving Eq. 8 is however not trivial due to its mixed set of discrete/continuous variables. To sidestep this intractability, we instead solve Eq. 8 via alternating between (1) optimizing $(\gamma, \zeta, \Omega)$ while fixing $b$; and (2) optimizing $b$ given $(\gamma, \zeta, \Omega)$. The first optimization sub-problem is straightforward as it reduces to derivations from Eq. 5, while the latter is less trivial due to the discrete nature of the optimizing variables $b$. Fortunately, we must recall that $b_m$ is exactly one-hot vector. This constraint is important in the condensation settings because it allows us to recast the non-linear log probability function to a linear form that can be solved effectively, as shown in Lemma E.1.

Given $(\gamma, \zeta, \Omega)$, this observation allows us to derive the linear form of Eq. 5 and Eq. 7 as direct consequences (see Appendix E for details), which reformulates Eq. 8 as follows:

$$b^* = \arg\max_b \left\{ \sum_{m=1}^M \log \mathbb{P}(z_m, b_m \mid \gamma, \zeta, \Omega) \right\} = \arg\max_b \left\{ R_1(b) + R_2(b) \right\}$$

$$\text{where:} \quad R_1(b) = \sum_{m=1}^M \sum_{k=1}^K b_{mk} \cdot \log \mathbb{N}(z_m \mid \omega_k; \text{diag}(\delta(\omega_k; \gamma))) \quad (9)$$

$$R_2(b) = \sum_{m=1}^M \sum_{k=1}^K b_{mk} \cdot \log \left( \frac{\exp(\alpha(\omega_k; \zeta))}{\sum_k \exp(\alpha(\omega_k; \zeta))} \right)$$

which is a weighted linear optimization task. Here, $R_1(b)$ is derived from log likelihood function of our pattern prior (see Eq. 5) while $R_2(b)$ originates from the log likelihood of our categorical assignment prior (see Eq. 7) using results of Lemma E.1. Here, we emphasize the many-to-one nature of the condensation problem, i.e., many patterns can be assigned to a single concept, by iteratively optimizing each $b_m$ while holding the remaining assignments $b_{-m}$ fixed. This reduces Eq. 9 to a maxima search problem that can be solved with linear complexity $\mathcal{O}(M)$. While solving Eq. 9, concepts that do not contribute to generating any observed patterns are treated as redundant and removed. This pruning mechanism gives NICER its nonparametric nature, enabling it to adapt to the varying complexity levels of WSIs. Pseudocode for NICER can be found in Appendix B

Table 1: Performance of baselines on Condensation Ability tasks. The best and second-best results are highlighted in **bold red**, and blue, respectively.

| Method | Decoder | Cancer Subtyping | | | | | | Survival Prediction | |
| | | PANDA | | | NSCLC | | | LUAD | BRCA |
| | | Kappa | Accuracy | F1 | Kappa | Bal. Acc. | F1 | C-Index | C-Index |
|---|---|---|---|---|---|---|---|---|---|
| Whole Bag | ABMIL | 91.93 ± 0.48 | 76.21 ± 1.53 | 76.37 ± 1.38 | 90.31 ± 1.65 | 94.52 ± 1.19 | 95.19 ± 0.78 | 62.12 ± 1.27 | 78.52 ± 3.82 |
| DeepSets | | 57.26 ± 38.13 | 51.60 ± 17.98 | 46.42 ± 23.91 | 79.51 ± 1.80 | 89.82 ± 0.88 | 89.73 ± 0.91 | 59.89 ± 5.34 | 49.23 ± 3.59 |
| ProtoCount | | 0.83 ± 9.55 | 24.24 ± 1.21 | 11.77 ± 1.28 | 10.69 ± 3.54 | 55.34 ± 1.78 | 53.86 ± 3.15 | 51.91 ± 5.75 | 56.47 ± 12.03 |
| H2T | | 75.03 ± 1.08 | 53.91 ± 1.10 | 50.66 ± 1.01 | 79.45 ± 1.80 | 89.67 ± 0.88 | 89.72 ± 0.90 | 51.83 ± 2.59 | 45.86 ± 4.29 |
| OT | | 41.92 ± 11.16 | 35.50 ± 3.22 | 29.98 ± 4.49 | 80.79 ± 4.69 | 90.39 ± 2.36 | 90.37 ± 2.37 | 54.07 ± 3.71 | 66.61 ± 6.30 |
| InfiniteGPFA | | 0.00 ± 0.00 | 26.85 ± 0.00 | 11.36 ± 0.00 | 6.61 ± 2.49 | 53.29 ± 1.25 | 50.61 ± 3.93 | 50.48 ± 4.35 | 46.65 ± 10.52 |
| PANTHER | | 65.52 ± 12.62 | 43.77 ± 0.76 | 42.52 ± 1.32 | 83.98 ± 2.39 | 92.02 ± 1.18 | 92.01 ± 1.21 | 58.92 ± 2.21 | 72.07 ± 6.17 |
| **NICER (Ours)** | | **90.98 ± 0.97** | **73.52 ± 1.27** | **73.22 ± 1.55** | **89.73 ± 1.81** | **94.84 ± 0.92** | **94.87 ± 0.91** | **64.67 ± 2.11** | **75.69 ± 2.02** |
| Whole Bag | DSMIL | 90.93 ± 0.43 | 74.05 ± 1.38 | 74.11 ± 1.31 | 84.60 ± 4.16 | 92.27 ± 2.08 | 92.30 ± 2.08 | 63.32 ± 4.02 | 67.96 ± 7.12 |
| DeepSets | | 85.53 ± 3.36 | 62.26 ± 1.77 | 62.88 ± 1.73 | 25.30 ± 36.76 | 62.68 ± 18.46 | 59.10 ± 20.77 | 56.86 ± 6.41 | 48.47 ± 15.91 |
| ProtoCount | | 14.78 ± 8.52 | 22.52 ± 2.11 | 14.47 ± 0.53 | -0.90 ± 13.16 | 49.52 ± 6.63 | 44.24 ± 10.37 | 52.71 ± 4.47 | 41.13 ± 8.69 |
| H2T | | 60.56 ± 1.57 | 41.83 ± 0.84 | 34.73 ± 5.23 | 44.90 ± 3.95 | 72.61 ± 2.03 | 70.10 ± 2.64 | 50.00 ± 0.00 | 49.82 ± 0.26 |
| OT | | 55.06 ± 7.77 | 37.66 ± 3.74 | 36.60 ± 3.94 | 48.26 ± 8.11 | 74.20 ± 4.00 | 73.52 ± 4.49 | 58.17 ± 4.11 | 70.84 ± 0.23 |
| InfiniteGPFA | | 17.72 ± 9.52 | 24.76 ± 2.19 | 20.56 ± 2.03 | -2.70 ± 12.64 | 48.62 ± 6.31 | 46.52 ± 7.29 | 46.25 ± 9.89 | 46.16 ± 6.69 |
| PANTHER | | 71.74 ± 2.46 | 46.09 ± 3.02 | 46.32 ± 3.28 | 75.05 ± 8.12 | 87.60 ± 4.04 | 87.46 ± 4.13 | 54.15 ± 4.88 | 69.96 ± 2.49 |
| **NICER (Ours)** | | **87.81 ± 0.69** | **66.07 ± 1.73** | **65.19 ± 1.65** | **84.61 ± 1.56** | **92.30 ± 0.74** | **91.34 ± 0.79** | **62.12 ± 1.41** | **74.53 ± 1.73** |
| Whole Bag | ILRA | 93.46 ± 0.24 | 78.90 ± 1.76 | 79.26 ± 2.05 | 87.17 ± 0.90 | 93.57 ± 0.41 | 93.58 ± 0.45 | 67.22 ± 2.12 | 81.46 ± 1.46 |
| DeepSets | | 75.49 ± 2.64 | 58.31 ± 1.12 | 54.52 ± 2.01 | 28.20 ± 39.89 | 64.10 ± 19.94 | 53.03 ± 27.79 | 56.72 ± 5.95 | 48.47 ± 15.91 |
| ProtoCount | | 8.65 ± 2.83 | 25.88 ± 0.83 | 14.98 ± 2.48 | 13.26 ± 8.39 | 56.63 ± 4.19 | 51.57 ± 10.17 | 55.02 ± 12.35 | 58.83 ± 6.48 |
| H2T | | 35.20 ± 17.79 | 35.05 ± 5.32 | 30.69 ± 6.22 | 77.50 ± 3.31 | 88.65 ± 1.69 | 88.69 ± 1.71 | 51.28 ± 1.81 | 49.82 ± 0.26 |
| OT | | 39.79 ± 7.28 | 37.73 ± 3.27 | 31.92 ± 1.93 | 83.98 ± 0.89 | 92.02 ± 0.41 | 91.98 ± 0.46 | 63.16 ± 3.32 | 62.86 ± 7.46 |
| InfiniteGPFA | | 0.62 ± 2.38 | 27.00 ± 0.56 | 13.28 ± 2.61 | 4.50 ± 5.49 | 52.26 ± 2.77 | 45.92 ± 8.34 | 51.67 ± 5.47 | 57.03 ± 8.21 |
| PANTHER | | 69.08 ± 3.63 | 48.62 ± 5.28 | 47.91 ± 5.45 | 81.41 ± 2.38 | 90.72 ± 1.14 | 90.69 ± 1.21 | 63.00 ± 9.98 | 69.18 ± 1.35 |
| **NICER (Ours)** | | **88.92 ± 2.37** | **69.73 ± 3.02** | **69.56 ± 3.59** | **88.47 ± 1.56** | **94.28 ± 0.76** | **94.23 ± 0.79** | **68.06 ± 4.69** | **76.99 ± 3.95** |

## 3 EXPERIMENTS

This section empirically evaluates NICER on four datasets across two key pathology tasks: cancer subtyping and survival prediction. Dataset details and baselines are provided in Sec. 3.1, while experimental results are reported in Sec. 3.2.

### 3.1 EXPERIMENT SETTINGS

**Datasets and Evaluation Metrics.** For cancer subtyping, we evaluate NICER on two different tasks: NSCLC subtyping on TCGA (2 classes), and ISUP grading based on PANDA challange (6 classes) (Bulten et al., 2022; 2020). In survival prediction, we evaluate NICER on TCGA across two cancer repositories: BRCA and LUAD. Following prior work (Song et al., 2024), we evaluate the cancer subtyping tasks using Cohen's Kappa (Vieira et al., 2010), accuracy, and weighted F1, replacing accuracy with balanced accuracy for NSCLC due to class imbalance. For survival tasks, we report the concordance index (C-Index) (Alabdallah et al., 2024).

**Evaluation Settings.** We view unsupervised prototype construction as both a condensation framework and a representation learning approach, and evaluate along two axes: (i) condensation ability, by applying NICER and baselines on the training set and testing against the original WSI feature bag; and (ii) morphological prototyping, by applying methods to both training and test sets.

**Baselines.** We consider two baseline categories: 1) *unsupervised prototyping* methods, which learn unsupervised representations followed by a task-specific neural predictor, and 2) *MIL-based predictors*, which construct supervised slide-level prototypical representations. For *unsupervised prototyping*, we compare NICER with **DeepSets** (Zaheer et al., 2017), **ProtoCounts** (Claudio Quiros et al., 2024), **H2T** (Yu et al., 2023), **InfiniteGPFA** (Yu et al., 2025), **Optimal Transport (OT)**(Mialon et al., 2021), and **PANTHER**(Song et al., 2024). Specifically, DeepSets, ProtoCounts, and H2T build prototypes from histological information and distance-based clustering (e.g., K-Means); InfiniteGPFA is adapted to perform latent factor analysis on each WSI; and OT and PANTHER adopt Gaussian Mixture Models for soft prototypical assignment. For *MIL-based predictors*, we adopt three supervised baselines: attention-based MIL (ABMIL) (Ilse et al., 2018), dual-stream MIL (DSMIL) (Li et al., 2021a), and low-rank MIL (ILRA) (Xiang & Zhang, 2023). These models are trained on unsupervised prototypes and evaluated on original feature bags to measure condensation ability, or used directly to assess NICER's effectiveness in producing slide-level unsupervised representations. Further implementation details are provided in Appendix C.

Table 2: Performance of baselines on Morphological Prototyping tasks. "TrL"/"TrM" is a linear/nonlinear transformer. The best and second-best results are highlighted in **bold red**, and blue, respectively.

| Method | | Cancer Subtyping | | | | Survival Prediction | |
|---|---|---|---|---|---|---|---|
| | | PANDA | | NSCLC | | LUAD | BRCA |
| | | Accur. | F1 | Bal. Acc. | F1 | C-Index | C-Index |
| MIL | ABMIL | 74.05 | 74.42 | 94.19 | 94.23 | 62.20 | 77.35 |
| | DSMIL | 72.48 | 72.52 | **95.17** | **95.19** | 68.90 | 77.72 |
| | ILRA | **76.96** | 77.11 | 93.21 | 93.26 | 55.26 | **83.43** |
| Unsupervised | DeepSets | 61.52 | 60.51 | 86.50 | 86.53 | 55.02 | 67.40 |
| | ProtoCount | 27.52 | 20.84 | 55.55 | 47.03 | 52.87 | 59.12 |
| | H2T | 55.93 | 53.81 | 77.75 | 77.75 | 53.83 | 52.85 |
| | OT | 73.15 | 72.87 | 88.42 | 88.45 | 64.59 | 75.51 |
| | InfiniteGPFA | 14.09 | 3.48 | 50.00 | 34.41 | 50.00 | 50.00 |
| | PANTHER | 70.47 | 69.98 | 82.69 | 82.68 | 45.45 | 75.51 |
| | PANTHER$_{TrM}$ | 70.02 | 70.06 | 88.53 | 89.45 | 60.29 | 67.59 |
| | **NICER$_{TrL}$** | 72.48 | 72.65 | 94.27 | 94.23 | 70.57 | 76.98 |
| | **NICER$_{TrM}$** | **76.96** | **77.12** | 95.17 | 95.19 | **70.81** | 81.40 |

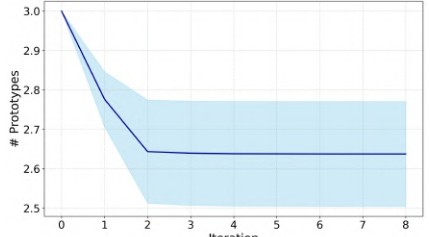

Figure 5: Number of prototypes over first 8 iterations on NSCLC dataset. The values are reported at $\log_{10}$ scale

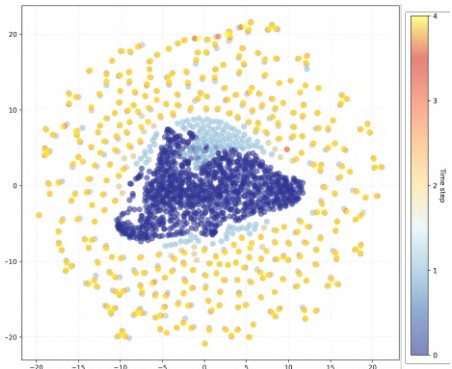

Figure 6: NICER concepts disperse over iterations. This shows that our concept prototypes encode different information from the WSI, ensure diversity.

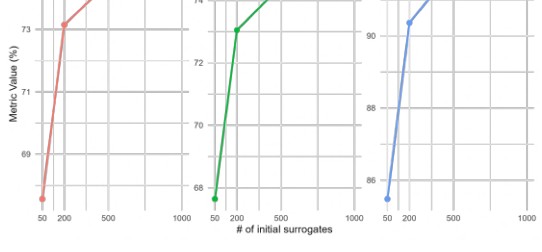

Figure 4: Impact of initial patterns number on NICER's overall performance.

## 3.2 PERFORMANCE ON CONDENSATION ABILITY

**Results on cancer subtyping.** Tab. 1 highlights that NICER consistently has superior performance across different predictors for cancer subtyping. On the PANDA dataset, when paired with complex predictors such as DSMIL and ILRA, NICER achieves improvements of 2.28–15.04% across evaluation metrics, with a particularly notable gain of 15.04% in F1 score over DeepSets, when using DSMIL. This advantage becomes even more pronounced with simpler architectures such as ABMIL, where the performance margin widens to 33.72% relative to the next-best baseline. Importantly, NICER sustains strong performance with only a minor drop from Whole Bag (∼2.87% in Kappa), offering a favorable trade-off between accuracy and efficiency. Consistent patterns are observed on the NSCLC dataset, where NICER surpasses competing methods by up to 5.27% in balanced accuracy. This is due to NICER's nonparametric design, which condenses WSIs based on their complexity, yielding robust representations across architectures.

**Results on survival prediction.** In survial prediction tasks, NICER consistently outperforms other baselines across diverse predictor architectures, achieving performance gains of up to 79.73%. On BRCA, NICER improves the C-Index by as much as 33.40% over baselines, with an average margin of 4.98% compared to the closest competitors. On LUAD, similar improvements are observed, with margins of 2.82% and 9.56% when combined with ABMIL and DSMIL predictors, respectively. Remarkably, when paired with the more complex ILRA predictor, NICER, along with methods such as PANTHER and ProtoCount, can surpass the Whole Bag upper bound, indicating that unsupervised feature construction can denoise raw feature bags and enhance downstream predictive performance. These results highlight both the robust effectiveness of NICER and its model-agnostic generality, underscoring its potential as a broadly applicable framework for histological analysis.

## 3.3 PERFORMANCE ON MORPHOLOGICAL PROTOTYPING

As shown in Tab. 2, NICER combined with a TrM consistently outperforms other unsupervised methods across all scenarios, demonstrating its ability to preserve semantic information from the

Table 3: Performance-Efficiency trade-offs comparison of NICER and the SOTA method, on PANDA using ABMIL.

| Method | Config | % Input | F1 |
|---|---|---|---|
| Whole Bag (Upperbound) | - | 100% | 74.42 |
| PANTHER | K=16 | 2.58% | 41.11 |
| | K=64 | 10.39% | 55.93 |
| | K=128 | 20.78% | 69.45 |
| NICER (Ours) | M=50 | 5.79% | 67.64 |
| | M=200 | 11.32% | 74.46 |
| | M=500 | 11.56% | 74.50 |
| | M=1000 | 11.31% | 75.00 |

Table 4: Impact of top-$\kappa$ in pattern distillation on NICER's performance. Conducted on PANDA with ABMIL predictor

| Task | $\kappa$ | Kappa | Acc. | F1 |
|---|---|---|---|---|
| Conden-sation | 1 | 90.18 | 70.02 | 69.87 |
| | 3 | 91.48 | 74.94 | 75.00 |
| | 5 | 90.47 | 72.93 | 72.72 |
| | 10 | 89.52 | 70.47 | 70.49 |
| Prototyping | 1 | 92.89 | 75.17 | 75.47 |
| | 3 | 93.07 | 77.85 | 77.96 |
| | 5 | 92.77 | 75.62 | 75.80 |
| | 10 | 92.89 | 73.83 | 73.82 |

original noisy feature bag. Due to its nonparametric learning process and decomposition of diversity and compactness, NICER constructs representations that achieve competitive performance, exhibit only minimal drops across diverse tasks and datasets, underscoring its model-agnostic nature. In contrast, the performance of conventional MIL methods is strongly dependent on the chosen architecture.

## 3.4 ABLATION STUDIES

**Impact of Initial Number of Patterns.** Fig. 4 illustrates the effect of the number of patterns initialized at the start of the NICER algorithm on feature construction performance. Experiments are conducted on the PANDA dataset and evaluated on the Condensation Ability task using ABMIL as the predictor. As shown, NICER achieves substantial performance gains as the number of patterns increases, but the improvement begins to saturate beyond a certain point (e.g., $\sim 200$). This trend arises because a larger pattern set provides greater distillation capacity, while the plateau reflects the condensation process, which removes redundancy and converges toward stable representations.

**Performance-Efficiency Trade-offs Comparison.** To assess the effectiveness of NICER in the condensation problem, we compare its performance-efficiency trade-off against PANTHER on the cancer subtyping task using the PANDA dataset across varying condensation levels. The condensation level is controlled by the number of prototypes $K$ in PANTHER and the number of initial patterns $M$ in NICER, where $M$ serves as an upper bound on the number of final concepts. As reported in Tab. 3, both methods exhibit an upward trend in F1 as the % Input increases. Notably, at comparable condensation levels (e.g., PANTHER with 10.39% Input vs. NICER with 11.32% Input), NICER surpasses PANTHER by a substantial margin of nearly 19%, underscoring its superior ability to preserve relevant information during condensation.

**Effectiveness of Condensation Stage.** Tab. 3 shows how $M$ influences the capacity of the condensed prototype set. With small $M$ (e.g., 50), the final set is limited by the initial pattern pool. As $M$ grows (e.g., beyond 200), the number of prototypes increases but quickly stabilizes, as seen in the (% Input) column. Downstream performance follows the same trend, with F1 scores plateauing once stabilization occurs. This confirms that NICER's condensation stage effectively merges redundant patterns into a compact yet informative concept set.

**Impact of Top-$\kappa$ in Pattern Learning.** We perform a sensitivity analysis on the number of patterns selected per patch feature (top-$\kappa$) across both Condensation Ability and Morphological Prototyping tasks. As shown in Tab. 4, performance steadily improves across all quantitative metrics as $\kappa$ increases, but begins to plateau or even slightly decline beyond a certain point (e.g., $\kappa = 3$). This occurs because excessively large $\kappa$ values cause each patch's information to be distributed too broadly across patterns, thereby reducing the effectiveness of the distillation process. These findings are consistent with our earlier observation in Fig. 4 and align with the design discussed in Sec. 2.3.

**Concept Prototypes Diversity.** To analyze the behavior of our condensation process, which maps patterns into concepts, we visualize 2D t-SNE embeddings of the concept set on the NSCLC dataset over the first five iterations. As shown in Fig. 6, the learned concepts become increasingly dispersed (brighter points) as training progresses, reflecting convergence toward diverse and specific infor-

mation. This demonstrates the effectiveness of the condensation process in reducing overlap and redundancy while preserving diversity, a key strength of NICER. More experimental results can be found in Appendix F

## 4 RELATED WORK

**Multiple Instance Learning.** While initial histology-based outcome prediction was centered on pathologist-annotated region-of-interests (Bychkov et al., 2018; Kather et al., 2019; Mobadersany et al., 2018), later works have utilized WSIs for clinical prediction tasks with MIL (Campanella et al., 2019; Chen et al., 2022; Tang et al., 2023; Nguyen et al., 2025b;a). There is a sustained effort for new MIL schemes, with a focus on developing new patch aggregation strategies to learn more representative and task-specific embedding, towards better predictive accuracy (Li et al., 2021b; Lu et al., 2023; Shao et al., 2021; Tang et al., 2023; Xiang & Zhang, 2023) or interpretability (Javed et al., 2022; Thandiackal et al., 2022). Recent MIL proposals further enhance efficiency during training and inference by adopting low-rank property of histological images (Xiang & Zhang, 2023) or using sparse coding model as a regularization in an attention-based aggregator (Qiu et al., 2023). NICER is similar to MIL in that the patch features in each WSI (represented as a bag) is aggregated and condensed in different ways to produce a slide-level embedding. Nevertheless, NICER performs in an unsupervised manner, in contrast to supervised MIL approaches.

**Prototype Learning.** Prototypes, representative examples summarizing datasets, have been widely used in bioinformatics and NLP (dan Guo et al., 2022; Kim, 2022; Lee et al., 2019; Mialon et al., 2021), appear under related notions such as signatures (Lazebnik et al., 2005; Zhang et al., 2006; Caicedo et al., 2009) and bag-of-visual-words (Caicedo et al., 2009; Cruz-Roa et al., 2009; Sivic & Zisserman, 2003). In computational pathology, prototypical representations are natural since repeating histology patterns often reflect shared morphology (Hou et al., 2016; Kalra et al., 2020; Pan et al., 2023; Wang et al., 2022b; Xu et al., 2012; Yu et al., 2023). Recent approaches (Vu et al., 2023; Claudio Quiros et al., 2024; Zaheer et al., 2017) build WSI prototypes using manual features and distance-based clustering (e.g., K-Means), with state-of-the-art variants adopting Gaussian Mixture Models (Mialon et al., 2021; Song et al., 2024). However, they all impose a fixed number of prototypes, ignoring the varying complexity of different WSIs. Adaptive clustering methods (Li & Nehorai, 2018; Vijayan & Aziz, 2023) offer partial flexibility but rely on rigid structures and incur high training costs, making them impractical for gigapixel slides. These limitations motivate NICER, a probabilistic nonparametric framework that is efficient and slide-adaptive.

**Dataset Condensation.** Dataset condensation, or distillation, compresses large datasets into small synthetic sets that preserve model performance (Wang et al., 2020). Unlike prototype learning, which selects subsets or analytic representations, it treats synthetic samples as learnable parameters and tries to balance between performance and efficiency via a bi-level learning approaches. This line of research typically involves in the nested optimization (Wang et al., 2020; Deng & Russakovsky, 2022; Nguyen et al., 2021); or surrogate-objective approaches (Zhao et al., 2021; Wang et al., 2022a; Liu et al., 2023; Sajedi et al., 2023). In the context of WSIs, FedWSIDD (Jin et al., 2025) extends condensation to federated settings by synthesizing slides for efficient communication, but like conventional methods it relies on supervision signals, limiting its use in real-world scenarios where annotations are scarce. In contrast, NICER introduces an unsupervised data condensation framework that eliminates the dependency on labels, enabling scalable, annotation-free WSI condensation.

## 5 CONCLUSIONS

In this paper, we introduce NICER, a novel framework for whole-slide image (WSI) condensation that addresses histological heterogeneity across slides. NICER reformulates prototype construction as an unsupervised data condensation problem within a hierarchical probabilistic model, where prototypical information is distilled from raw features into patterns and then condensed into compact concepts nonparametrically. This adaptive process allows the prototype set to scale with WSI complexity, effectively handling variability across slides. We further derive a Bayesian inference algorithm to learn pattern-concept associations efficiently. Across datasets and tasks, NICER outperforms prior methods with up to 90% gains and strong efficiency, demonstrating practicality for real-world pathology under limited resources.

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

## A    BROADER STATEMENT OF IMPACT

This research develops an effective nonparametric compression and condensation algorithm for whole-slide images (WSIs), enabling efficient learning from histological data with varying complexity. The mathematical methods and insights presented in this work help bridge the gap between large, gigabyte-scale images and their practical applications in healthcare. While the potential application of our methods to real patient data may raise ethical considerations, such effects are indirect and not the focus of this study. Our experiments rely solely on publicly available datasets, ensuring that no ethical concerns are introduced in the evaluation of our algorithms.

## B    PSEUDOCODE FOR NICER

---
**Algorithm 1** Nonparametric Unsupervised Data Condensation (NICER)

---
**input**: WSI feature bag $\mathcal{H}$, no. $T$ of iterations, no. $M$ of initial patterns
**output**: condensed concept prototype set $\Omega$

1: initialize pattern set $\mathcal{Z}$ of size $M$
2: **for** $t = 1$ **to** $T$ **do**
3:     **for** $i = 1$ **to** $N$ **do**
4:         query top-$\kappa$ relevant patterns to $h_i$ using Eq. 3
5:         distill $h_i$ to $\kappa$ relevant $z_m$ by maximizing Eq. 3
6:     **end for**
7:     $\Omega, b \leftarrow \text{condense}(\{z_m\}_{m=1}^{M})$ // solving Eq. 8
8:     $\Omega \leftarrow \left\{ \omega_k \in \Omega \mid \sum_{m=1}^{M} b_{mk} > 0 \right\}$ // remove redundant concepts
9: **end for**
10: **return** the set $\Omega$ of optimal concepts

---

## C    IMPLEMENTATION DETAILS

### C.1    DATASETS

We provide brief explanations for the datasets that were used for the evaluation of NICER for condensation ability and prototyping ability.

**PANDA.** (Bulten et al., 2022; 2020) For the ISUP grading task, we used prostate cancer core needle biopsies (n=10,616) from the Prostate Cancer Grade Assessment (PANDA) challenge. Each biopsy is given an ISUP grade, making this a 6-class classification task. These biopsies are collected from Karolinska Institute (KRLS) and Radboud University Medical Center (RUMC). We label-stratify the PANDA dataset into train/val/test of 80:10:10 and performance was evaluted using Cohen's quadratic weighted Kappa $\kappa^2$, accuracy and weighted F1 metrics.

**NSCLC.** For the non-small cell lung carcinoma (NSCLC) subtyping task, we use H&E WSIs from TCGA for classifying lung adenocarcinoma (LUAD) and lung squamous cell carcinoma (LUSC) cases. The TCGA cohort contains a total of 1,4041 slides (LUAD: 529, LUSC: 512). We label-stratify the TCGA cohort into train/val/test fold of 80:10:10 and evaluate performance using Cohen's Kappa, balanced accuracy and F1 metrics.

**TCGA-BRCA.** The Breast Invasive Carcinoma (BRCA) cohort from The Cancer Genome Atlas (TCGA). a joint effort of the NCI and NHGRI, provides one of the most comprehensive digital pathology resources for breast cancer. The dataset contains 1,133 diagnostic WSIs spanning 1,062 patients, covering diverse histological and molecular subtypes. In addition to imaging, the cohort supplies curated clinical outcomes, including Overall Survival (OS) and Progression-Free Interval (PFI), which have been widely adopted as endpoints for developing and benchmarking survival prediction models. We label-stratify the TCGA-BRCA dataset into train/val/test fold of 80:10:10 and evaluate performance C-Index for survival prediction

**TCGA-LUAD.** The Lung Adenocarcinoma (LUAD) cohort within TCGA offers a large-scale, multi-institutional collection of pathology images and outcome data for a major subtype of non-

small cell lung cancer. It includes 529 diagnostic WSIs corresponding to 478 patients, with extensive clinical annotation. As with BRCA, the LUAD cohort provides OS and PFI as standardized survival endpoints, enabling robust prognostic modeling and cross-study comparison in computational pathology research. We label-stratify the TCGA-LUAD dataset into train/val/test fold of 80:10:10 and evaluate performance C-Index for survival prediction

## C.2 EVALUATION SETTING DETAILS

We view unsupervised prototype construction not only as a condensation framework, whose primary role is to reduce redundancy in the WSI feature bag, but also as a form of representation learning, since the resulting prototypes are later used as inputs for downstream predictors. Accordingly, we evaluate all methods along two complementary axes:

**Condensation ability.** Here, the goal is to assess how well the condensed prototype set preserves information from the original WSI feature bag. Specifically, we apply NICER and baseline methods on the training set to obtain prototypes, and then evaluate them on the original uncompressed bag of features. This setup isolates the effectiveness of condensation by measuring how much predictive power is retained (or lost) after summarization, independent of downstream task complexity. It answers the key question: *Does condensation discard critical information or faithfully represent the original slide?*

**Morphological prototyping.** In this setting, we follow the prior evaluation protocol Song et al. (2024) to test whether condensed prototypes can generalize as useful, task-agnostic representations. Condensation methods are applied to both the training and test sets, and the resulting prototypes are directly used for downstream prediction. Unlike the condensation ability evaluation, this setup emphasizes the representation learning capacity of the prototypes, focusing on whether they capture robust morphological cues transferable across unseen WSIs rather than reconstruction fidelity. To ensure fairness, we use linear probing for all baselines, thereby isolating the quality of the learned prototypes. For PANTHER and NICER, we additionally evaluate with a transformer layer (with or without linearity), since NICER produces prototype sets whose capacity adapts to WSI complexity and thus cannot be fully exploited by a fixed linear layer alone.

## C.3 TRAINING DETAILS AND COMPUTATIONAL RESOURCES

**Data preparation.** WSIs at $20\times$ magnification ($0.5, \mu m$/pixel) are divided into non-overlapping $256 \times 256$ patches, and all patches are used without sampling. These patches are converted to representations using UNI Chen et al. (2024), a pretrained foundation encoder. We set $\kappa = 3$ and found $T = 50$ iterations sufficient for convergence across all datasets.

**Hyperparameter settings.** For training, we adopt the AdamW optimizer with weight decay set to $1 \times 10^{-5}$ and employ a cosine decay learning rate scheduler. In the *cancer subtyping* experiments, models are trained with cross-entropy loss for up to 50 epochs, with early stopping triggered if the validation loss fails to improve for 10 consecutive epochs. The initial learning rate is set to $1 \times 10^{-4}$. Since both the original feature bags and the NICER's representations form variable-length WSI sets, we use a batch size of 1 combined with gradient accumulation over 32 steps across all methods. For the *survival prediction* task, we optimize using the negative log-likelihood (NLL) loss (Zadeh & Schmid, 2021), training over 50 epochs with a per-patient batch size of 1 and an initial learning rate of $1 \times 10^{-5}$. The training of NICER further involves an alternating optimization process over 20 iterations, with $\kappa = 3$ and an initial pattern set size of $M_0 = 1000$, which decreases progressively through the condensation procedure. As a nonparametric random process, NICER allows us to directly regulate the number of generated prototypes via $\kappa$ and $M_0$. Following prior practices (Vu et al., 2023; Song et al., 2024), we set a number of prototypes generated by unsupervised baselines to 16 for all WSIs used in our experiments. The implementation details of predictor architectures, unsupervised baselines follow original papers and previous settings (Song et al., 2024).

**Computational considerations.** All experiments and data preprocessing are conducted on a NVIDIA RTX A6000 with 46GB of memory.

## D   DERIVATION OF EQ. 2

We begin from the conditional likelihood in Eq. 2:

$$\log \mathbb{P}(\mathcal{H} \mid \mathcal{Z}, \theta) = \sum_{i=1}^{N} \log \mathbb{N}(h_i \mid z_{(i)}^*, \sigma^2 \mathbf{I}), \tag{10}$$

where each feature $h_i$ is modeled as a Gaussian centered at its assigned pattern prototype $z_{(i)}^*$. Expanding the Gaussian log-likelihood yields:

$$\log \mathbb{P}(\mathcal{H} \mid \mathcal{Z}, \theta) = -\frac{1}{2\sigma^2} \sum_{i=1}^{N} \|h_i - z_{(i)}^*\|^2 + C, \tag{11}$$

with $C$ denoting terms independent of $h_i$ or $z_{(i)}^*$. The squared distance can be written as

$$\|h_i - z_{(i)}^*\|^2 = \|h_i\|^2 + \|z_{(i)}^*\|^2 - 2\langle h_i, z_{(i)}^* \rangle. \tag{12}$$

Since both $h_i$ and $z_{(i)}^*$ are $\ell_2$-normalized embeddings, we have $\|h_i\|^2 = \|z_{(i)}^*\|^2 = 1$. This reduces the squared distance to

$$\|h_i - z_{(i)}^*\|^2 \approx 2 - 2\langle h_i, z_{(i)}^* \rangle. \tag{13}$$

Plugging this into the log-likelihood gives

$$\log \mathbb{P}(\mathcal{H} \mid \mathcal{Z}, \theta) \approx -\frac{1}{2\sigma^2} \sum_{i=1}^{N} \left( 2 - 2\langle h_i, z_{(i)}^* \rangle \right) + C. \tag{14}$$

Dropping constants, we recover Eq. 2. This shows that maximizing the Gaussian likelihood is approximately equivalent to maximizing feature - prototype similarity, providing a probabilistic justification for our design.

## E   LEMMAS AND DERIVATIONS

**Lemma E.1.** *(adapted from Weng et al. (2024) ) For any scalar function $g(\mathbf{r})$ and a binary vector $\xi = [\xi_1, \xi_2, \ldots, \xi_n]$ such that $\xi_i \in \{0, 1\}$ and $\xi$ has exactly one non-zero component, we have*

$$g\left( \sum_{i=1}^{n} \xi_i \cdot \mathbf{r}_i \right) = \sum_{i=1}^{n} \left( \xi_i \cdot g(\mathbf{r}_i) \right) \tag{15}$$

with respect to any set $\{\mathbf{r}_i\}_{i=1}^{n}$ of valid inputs to $g(\mathbf{r})$.

**Proof.** First, if there is no non-zero component, both sides of Eq. 23 evaluate to $g(0)$. Otherwise, suppose the only non-zero component appears at position $j$, both sides of Eq. 23 will evaluate to $g(\mathbf{r}_j)$. In both cases, Eq. 23 holds.

**Lemma E.2.** *Let $\mathbb{P}(z_m \mid b_m, \Omega$ defined as in Eq. 5. Let $R_1(b) \triangleq \sum_{m=1}^{M} \log \mathbb{P}(z_m \mid b_m, \Omega)$, considering $(z_m, \Omega)$ as constants. We have*

$$R_1(b) = \sum_{i=1}^{M} \sum_{k=1}^{K} b_{mk} \cdot \log \mathbb{N} \left( z_m \mid \omega_k, diag \left( \delta \left( \omega_k; \gamma \right) \right) \right), \tag{16}$$

*which is linear in terms of the assignment parameter b.*

**Proof.** To derive results of Lemma E.2, note that Eq. 5 implies the following,

$$\log \mathbb{P}(z_m \mid b_m, \Omega) = \log \mathbb{N} \left( z_m \mid \sum_{k=1}^{K} b_{mk} \cdot \omega_k, \text{diag} \left( \delta \left( \sum_{k=1}^{K} b_{mk} \cdot \omega_k; \zeta \right) \right) \right) \tag{17}$$

$$= g\left( \sum_{k=1}^{K} b_{mk} \cdot \omega_k \right) \tag{18}$$

where we define

$$g(\mathbf{x}) \triangleq \log \mathbb{N}\bigg( z_m \mid \mathbf{x}, \mathrm{diag}\,(\mathbf{x}; \zeta) \bigg) \tag{19}$$

In addition, since $\sum_k b_{mk} = 1$ with $b_{mk} \in \{0, 1\}$, Lemma E.1 implies that

$$g\bigg( \sum_{k=1}^{K} b_{mk} \cdot \omega_k \bigg) = \sum_{k=1}^{K} \bigg( b_{mk} \cdot g(\omega_k) \bigg) \tag{20}$$

We then plug Eq. 20 into Eq. 18 to have

$$\log \mathbb{P}(z_m \mid b_m, \Omega) = g\bigg( \sum_{k=1}^{K} b_{mk} \cdot \omega_k \bigg) = \sum_{k=1}^{K} \bigg( b_{mk} \cdot \omega_k \bigg) \tag{21}$$

$$= \sum_{k=1}^{K} b_{mk} \cdot \log \mathbb{N}\bigg( z_m \mid \omega_k, \mathrm{diag}\big(\sigma(\delta_k; \zeta)\big) \bigg) \tag{22}$$

Finally, taking summation over $m = 1, 2, \ldots, M$ on both sides of Eq. 22, we arrive Lemma E.2.

**Lemma E.3.** *Let $\mathbb{P}(b_m \mid \zeta)$ defined as in Eq. 7. Let $R_2(b) \triangleq \sum_{m=1}^{M} \log \mathbb{P}(b_m \mid \zeta)$, considering $(\zeta)$ as constants. We have*

$$R_2(b) = \sum_{i=1}^{M} \sum_{k=1}^{K} b_{mk} \cdot \log \left( \frac{\exp\big(\alpha(\omega_k; \zeta)\big)}{\sum_k \exp\big(\alpha(\omega_k; \zeta)\big)} \right), \tag{23}$$

*which is linear in terms of the assignment parameter b.*

**Proof.** Plug Eq. 7 into the definition of $R_2(b)$, we have

$$R_2(b) = \sum_{m=1}^{M} \log \mathbb{P}(b_m \mid \zeta) = \sum_{m=1}^{M} \log \left( \prod_{k=1}^{K} \left( \frac{\exp\big(\alpha(\omega_k; \zeta)\big)}{\sum_k \exp\big(\alpha(\omega_k; \zeta)\big)} \right)^{b_{mk}} \right) \tag{24}$$

$$= \sum_{m=1}^{M} \sum_{k=1}^{K} b_{mk} \cdot \log \left( \frac{\exp\big(\alpha(\omega_k; \zeta)\big)}{\sum_k \exp\big(\alpha(\omega_k; \zeta)\big)} \right) \tag{25}$$

which naturally arrives Lemma E.3.

# F  ADDITIONAL RESULTS

## F.1  PROTOTYPE ANALYSIS

**Concept Prototype Diversity and Convergence.**  We extend the findings of Section 3.2 by examining additional WSI instances from TCGA-NSCLC. Figure 7 illustrates the trajectories of concept prototypes over 10 optimization iterations of NICER for three representative samples: *TCGA-93-A4JN-01Z-00-DX1.ED4C9365-6CCF-4AEE-B4C9-3CC5EC57286C*, *TCGA-50-6594-01Z-00-DX1.43b2005a-4245-4025-ad85-4a957f308a5c*, and *TCGA-49-4514-01Z-00-DX2.f1565a36-257d-432e-a84d-47c1d7a0185f*. The visualizations reveal that different WSIs exhibit distinct condensation dynamics, reflecting variation in morphological complexity and feature distribution. For example, the first sample (ending in "86C") shows prototypes that initially cluster tightly, suggesting greater homogeneity, whereas the other two slides begin with more diffuse clusters, indicating higher heterogeneity.

Despite these sample-specific differences, a consistent pattern emerges across all trajectories: prototypes gradually diverge from their initialization with increasing variance, reflecting how condensation enforces specialization and reduces redundancy while preserving diversity. This behavior highlights NICER's ability to uncover distinct and non-overlapping concept structures within each slide. Complementary evidence is shown in Figure 10, where the number of prototypes stabilizes after only a few iterations. This rapid convergence indicates that redundant concepts are pruned early, leaving a compact and stable set that continues to refine qualitatively rather than quantitatively. Together, these results emphasize NICER's efficiency in learning diverse, non-redundant representations of WSIs with minimal optimization steps.

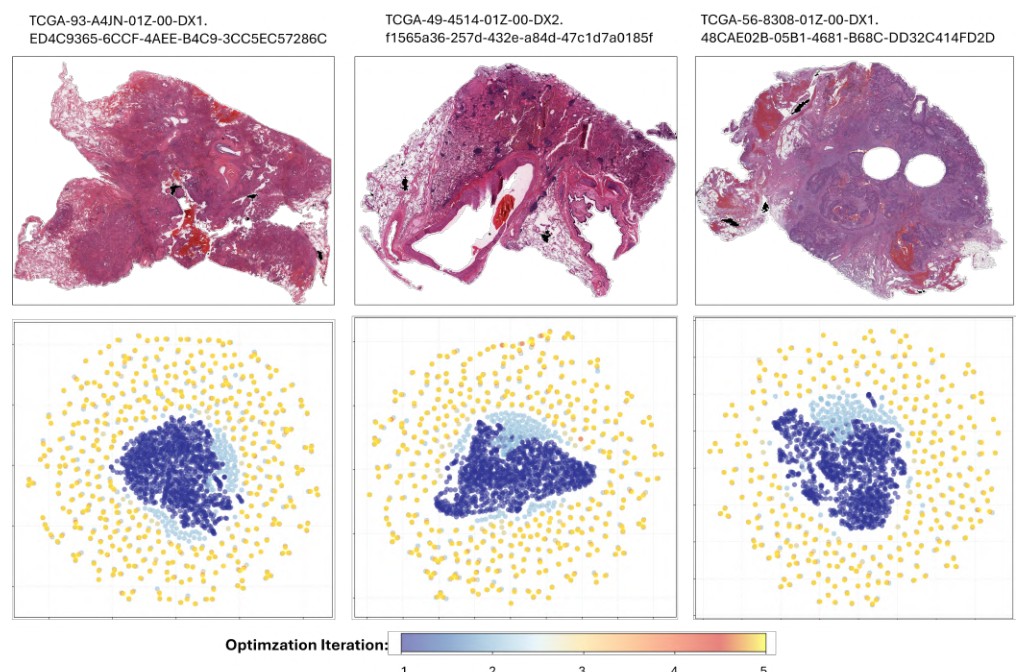

Figure 7: 2D t-SNE visualizations of NICER's concept prototypes learned over five iterations using three representative TCGA slides. These plots highlight that NICER can capture different diverse information across different WSIs.

Table 5: Average number of final concept prototypes across different datasets

| PANDA | NSCLC | LUAD (Survival) | BRCA (Survival) |
|-------|-------|-----------------|-----------------|
| 62.80 | 154.67 | 157.40 | 158.60 |

**Diversity Preservation.** To evaluate NICER's ability to preserve diversity across whole-slide images (WSIs), we compare 2D t-SNE visualizations of concept prototypes generated by NICER and PANTHER, alongside the original feature bags, on three TCGA-NSCLC samples: *TCGA-93-A4JN-01Z-00-DX1.ED4C9365-6CCF-4AEE-B4C9-3CC5EC57286C*, *TCGA-50-6594-01Z-00-DX1.43b2005a-4245-4025-ad85-4a957f308a5c*, and *TCGA-49-4514-01Z-00-DX2.f1565a36-257d-432e-a84d-47c1d7a0185f*. For fair visualization, NICER's prototypes are clustered with K-Means to 16, matching the fixed prototype count used in PANTHER (Song et al., 2024). As shown in Figure 8, PANTHER fails to capture the inherent diversity of WSIs, collapsing heterogeneous regions into a limited number of clusters and discarding critical information required for downstream tasks, an effect consistent with its suboptimal performances in Table 1 and Table 2. This limitation arises because PANTHER enforces a rigid and heuristically small prototype budget, prioritizing efficiency at the expense of representational fidelity. In contrast, NICER employs an alternating optimization strategy that leverages pattern-based diversity preservation and condensation-driven efficiency, ensuring prototypes remain well-separated and encode distinct, slide-specific information. This balance enables NICER to adapt to the variability of individual WSIs while maintaining compact yet expressive concept sets.

**Concept Prototype Capacity Analysis**. To evaluate NICER's ability to adapt its capacity to complexity of data, we report the average number of concept prototypes across different datasets. As can be seen from Table 5, for larger and more complex WSIs (e.g., NSCLC, LUAD, BRCA), NICER naturally allocates more capacity to preserve the underlying conceptual diversity in the slides. This aligns with our observation in Section 3.

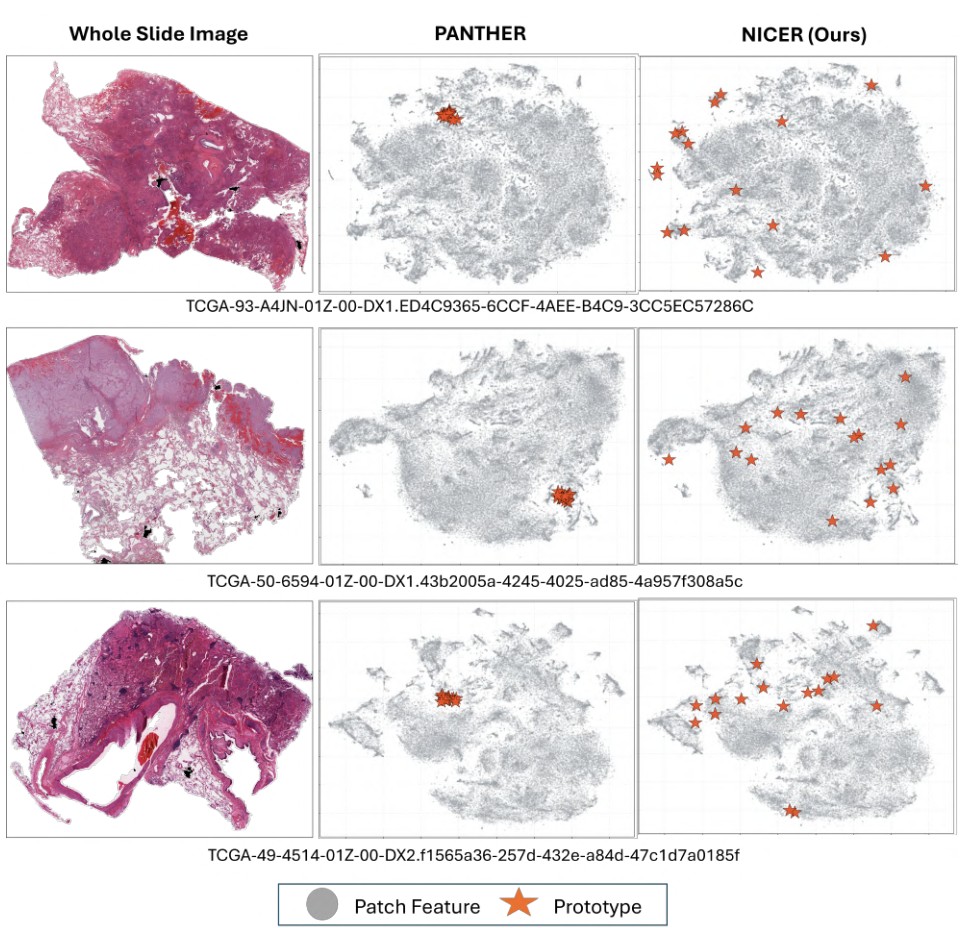

Figure 8: 2D t-SNE visualization of original WSI feature bags and learned concept prototypes, comparing our method (NICER) with a representative prior approach (PANTHER).

Table 6: Performance-Efficiency trade-offs comparison of NICER and the SOTA method. Conducted on PANDA using AB-MIL.

| Predictor | $M$ | Kappa | Accuracy | F1 |
|-----------|-----|-------|----------|-----|
| **ABMIL** | 50 | 85.48 | 67.56 | 67.64 |
|  | 200 | 90.36 | 73.15 | 73.04 |
|  | 500 | 91.85 | 74.50 | 74.46 |
|  | 1000 | 91.48 | 74.94 | 75.00 |
| **DSMIL** | 50 | 82.75 | 60.18 | 61.03 |
|  | 200 | 86.74 | 66.89 | 64.56 |
|  | 500 | 88.59 | 67.34 | 68.03 |
|  | 1000 | 88.73 | 68.46 | 67.52 |
| **ILRA** | 50 | 84.41 | 65.55 | 65.13 |
|  | 200 | 89.67 | 66.62 | 67.34 |
|  | 500 | 92.25 | 73.83 | 74.55 |
|  | 1000 | 92.25 | 73.83 | 74.55 |

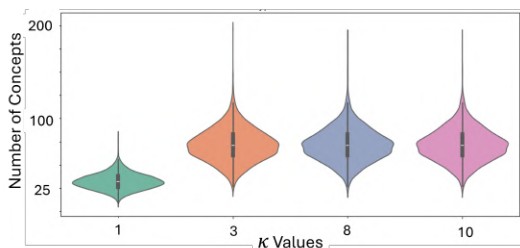

Figure 9: Impact of top-$\kappa$ in number of final concept prototypes, conducted on PANDA. The plots show that number of concepts increases when we increase $\kappa$, allowing more preservation capacity. After a certain point, the condensation process saturates this count to a stable value range.

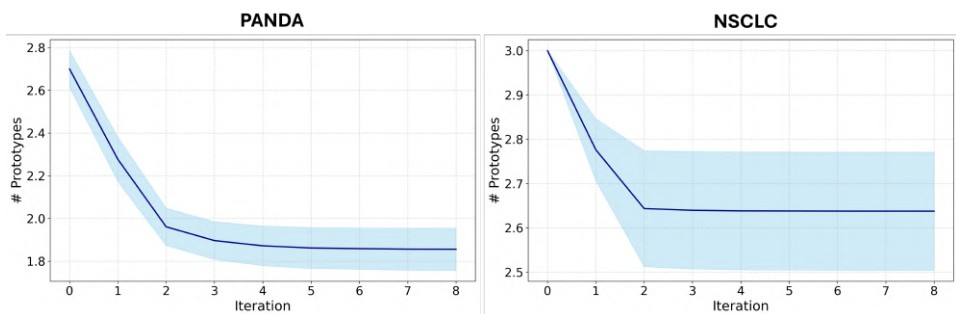

Figure 10: Number of prototypes tracked over the first 8 iterations on PANDA and NSCLC dataset. The values are reported at log10 scale

Table 7: Ablation study of top-$\kappa$ on downstream tasks performance across different predictor architectures. The experiments are conducted on PANDA dataset.

| Predictor | ABMIL | | | DSMIL | | | ILRA | | |
|---|---|---|---|---|---|---|---|---|---|
| Top-K | Kappa | Acc. | F1 | Kappa | Acc. | F1 | Kappa | Acc. | F1 |
| 1 | 90.18 | 70.02 | 69.87 | 85.42 | 64.65 | 65.95 | 89.17 | 64.88 | 64.24 |
| 3 | 91.48 | 74.94 | 75.00 | 88.73 | 68.46 | 67.52 | 92.25 | 73.83 | 74.55 |
| 5 | 90.47 | 72.93 | 72.72 | 88.80 | 68.46 | 66.66 | 88.66 | 71.36 | 71.47 |
| 10 | 89.52 | 70.47 | 70.49 | 85.42 | 66.89 | 66.94 | 88.88 | 68.90 | 68.92 |

## F.2 SENSITIVITY ANALYSIS

**Extensive Results on Impacts of Number of Initial Prototypes ($M$).** Table 6 complements the analysis in Figure 4 by evaluating how the initial number of patterns influences NICER's performance across different predictor architectures. The results demonstrate that increasing the pattern set size consistently improves performance, as a larger pool enhances the model's capacity to distill informative representations. However, these gains diminish once the pattern set grows beyond a certain threshold, with performance gradually reaching a plateau. This saturation reflects the role of the condensation process, which systematically eliminates redundancy and stabilizes the number of effective representations required to characterize a WSI, regardless of the starting pattern count, aligning our insights discussed from Figure 4.

**Extensive Results on Impacts of Top-$\kappa$ during Pattern Exploration.** We further conduct a sensitivity analysis on the number of pattern assigned to each patch feature, controlled by the top-$\kappa$ selection in the Condensation Ability tasks. The results, reported in Tab. 7, reveal a clear trend: model performance improves steadily across all evaluation metrics as $\kappa$ increases, indicating that incorporating multiple patterns per patch allows richer information to be preserved. However, beyond a moderate value (e.g., $\kappa = 3$), this benefit begins to diminish, with performance gains plateauing or even slightly decreasing. The degradation at larger $\kappa$ arises because information from each patch becomes overly dispersed across many patterns, weakening the sharpness of the distilled representation. These observations corroborate our earlier findings in Fig. 4 and Tab. 4, and reinforce the design principle outlined in Sec. 2.3 that pattern assignments must balance informativeness with compactness.

**Extensive Results on Effectiveness of Condensation Stage.** Fig. 9 shows the distribution of the number of final concepts produced by NICER across different $\kappa$ values. As expected, the average number of prototypes increases with larger $\kappa$, since greater capacity enables the model to capture more diverse patterns from complex WSIs. Beyond a certain point, however, this growth plateaus, indicating that the condensation process has effectively merged overlapping patterns and compressed them into a compact, stable concept set. This stabilization highlights NICER's ability to balance capacity with redundancy removal, yielding a consistent number of meaningful prototypes.

# G LIMITATIONS

Despite NICER's strengths in handling heterogeneous WSI complexity under limited-resource settings, there remain several avenues for improvement that we plan to explore in future work. First, NICER relies on a bag-of-features paradigm, where patches are treated independently and fine-grained spatial or multi-scale context is ignored. This prevents explicit modeling of tissue architecture and spatial priors—an important direction for future work, particularly in clinically critical settings. Second, prototype interpretability remains underexplored. While NICER's nonparametric concepts improve efficiency and performance, their clinical meaning and uncertainty calibration have not been systematically assessed. Bridging these gaps is essential for making condensed representations both effective and trustworthy in medical applications.

