# OpenReview forum: "Nonparametric Unsupervised Data Condensation for Gigapixel Histological Images"
_ICLR.cc/2026/Conference — ICLR 2026 Conference Withdrawn Submission_

### Official Review · Reviewer_SG2S · 2025-10-27

**Soundness:** 3
**Presentation:** 4
**Contribution:** 3
**Rating:** 6
**Confidence:** 4

**Summary:**

The paper describes NICER, a framework for unsupervised learning in histopathology images, based on non-probabilistic data condensation for reducing the number of prototypes through the definition of meta prototypes (or concepts).
In this way, authors formalize e new paradigm for histological representation learning, with the main claim of having a framework able to create adaptive sets of prototypes according to the slide. They provide extensive assessment and benchmark against recent baselines on a variety of pathology tasks, showcasing significant improvements.

**Strengths:**

1. Presentation is excellent: the way the framework is framed, the narrative flow, and the highly polished visuals makes the manuscript really enjoyable.
2. Relevance: the depth of theoretical framework, and the proposed ideas makes the manuscript particularly adequate for ICLR.
3. Novelty: as far as I could find in the literature, the idea of condensing and reducing prototypes has not yet applied in histopathology and it is particularly valuable. The theoretical framework is well conceived and gives additional strength to the story.
4. Technical soundness: the concepts are clearly explained, and most of details for replicability are provided, even if availability of source code would be preferrable.
5. Assessment and comparison: authors report on extensive comparisons against SOTA methods on a variety of tasks, and the results appear generally convincing.

**Weaknesses:**

The paper appears to be solid, but I have some concerns about the fairness in comparison, and the reporting of results:
1. It seems to me that $K$ is the most important parameter, since it represents the initial target of number of concepts, that is reduced during condensation. It would be important to also report on the final number of concepts after the pipeline is complete.
2. Qualitative results and visual explainability: Fig.6 and Fig.8. Why tSNE was chosen for performing projection? I think it would provide a bias and also artifacts, as I see some small weird triangular clusters after few iterations. How do you explain them?
Also, as far as I could understand, the final set of concepts are associated to visual features and they can be represented visually (for example for generating exemplar patches from histopathology slide). I would have expected that authors provided qualitative results about prototypical assignment, and examples on how these prototypes look, like it is done in PANTHER.
3. Fairness of comparison: as far as I could understand, the main competitor is PANTHER, but I don't think that the benchmarking is completely fair. For example, it is not clear what parameters and conditions are considered for creating Table 3; and also Figure 8 is not fair, since it seems that the prototypes obtained from Panther has limited expressivity, while instead from the original manuscript the prototypical assignment map shows the opposite; I think that this abrupt clustering of prototypes is a bias effect due to tSNE projection, but I do not feel it can be interpreted as limitation in expressivity power of the prototypes found with PANTHER framework.

**Questions:**

1. Minor: in the introduction, I would mention about tile-based processing that is inherently parallel.
Line 209: is. $\zeta$ parameteres appear out of nowhere in line 225. A table representing and explaining all parameters would help in readability of the manuscript.
2. The way the paradigm is theoretically framed reminds about sparse coding with overcomplete dictionaries:
Aharon, M., Elad, M., & Bruckstein, A. (2006). K-SVD: An algorithm for designing overcomplete dictionaries for sparse representation. IEEE Transactions on signal processing, 54(11), 4311-4322. I would like that authors try to analize analogies and differences.
3. Scalability performance: what do you expect as trade-off for condensation? How much does it cost to perform condensation instead of keeping the full WholeOfBag?

**Details Of Ethics Concerns:**

No concerns

---

> ### Author Response · Authors · 2025-11-24
>
> We thank the reviewer for reviewing our work and their constructive comments.
>
> __Q1. "It seems to me that K is the most important parameter, since it represents the initial target of number of concepts, that is reduced during condensation. It would be important to also report on the final number of concepts after the pipeline is complete."__
>
> Thank you for this helpful suggestion. In NICER, the capacity of the concept set is controlled through the upper bound $M$, and the nonparametric mechanism automatically determines the final number of concepts based on slide complexity. This design is important because WSIs are highly heterogeneous, making it difficult to predefine an appropriate target number of prototypes.
>
> As you noted, it is valuable to report the final number of concepts. In practice, we observe that the final concept count remains stable even when increasing the upper bound (M), as long as the slide complexity is fixed. We highlight this behavior in Table 3 and Figure 9 (with reported number of concepts), and will make this point more explicit in the revision.
>
> __Q2. "Qualitative results and visual explainability: Fig.6 and Fig.8. Why tSNE was chosen for performing projection? I think it would provide a bias and also artifacts, as I see some small weird triangular clusters after few iterations. How do you explain them? Also, as far as I could understand, the final set of concepts are associated to visual features and they can be represented visually (for example for generating exemplar patches from histopathology slide). I would have expected that authors provided qualitative results about prototypical assignment, and examples on how these prototypes look, like it is done in PANTHER."__
>
> __t-SNE Projection.__ We thank the reviewer for the constructive comments. t-SNE is a widely used and well-accepted tool for visualizing high-dimensional data distributions, despite its known limitations. To improve clarity, we additionally provide a UMAP version of Figure 6 in this link [UMAP image](https://github.com/anon-user-0/NICER-submission/blob/main/analysis_results/UMAP_NICER/umap_diversity.jpg), which shows that our qualitative conclusions remain unchanged across visualization methods.
>
> __Triangular Clusters.__ The triangular shapes observed in the t-SNE plot are likely artifacts of the projection and should not be overinterpreted. As shown in the UMAP visualization, our key claim - that the diversity of concept prototypes increases as training progresses - continues to hold, independent of such projection artifacts.
>
> __Biological visualization.__ We thank the reviewer for raising this point. Following PANTHER’s evaluation protocol, we visualize the patch-concept assignments and overlay them on the original WSIs (see this link [biological interpretation](https://github.com/anon-user-0/NICER-submission/blob/main/analysis_results/Biology_explain/bio.jpg )). These visualizations show that NICER produces clearer and more coherent region separation, indicating that NICER groups morphologically similar patches more faithfully. This aligns with established observations in the WSI literature that similar tissue structures tend to cluster spatially [1].
>
> [1] Ho et al., Deep Learning-Based Objective and Reproducible Osteosarcoma Chemotherapy Response Assessment and Outcome Prediction. Am J Pathol. 2023

---

> > ### Comment · Reviewer_SG2S · 2025-11-27
> > **Answer Q1 and Q2**
> >
> > Did you already update the table in the manuscript? I cannot find clear indication on what is the final number of concepts,
> > and the answer to my question does not appear clear. Please provide also preliminary results as official comment.
> > Also for Q2, I would expect that Fig. 8 is redone with UMAP, and for reference, please consider also PCA, or justify why to discard it.

---

> ### Author Response · Authors · 2025-11-24
>
> __Q3. "Fairness of comparison: as far as I could understand, the main competitor is PANTHER, but I don't think that the benchmarking is completely fair. For example, it is not clear what parameters and conditions are considered for creating Table 3; and also Figure 8 is not fair, since it seems that the prototypes obtained from Panther has limited expressivity, while instead from the original manuscript the prototypical assignment map shows the opposite; I think that this abrupt clustering of prototypes is a bias effect due to tSNE projection, but I do not feel it can be interpreted as limitation in expressivity power of the prototypes found with PANTHER framework."__
>
> We thank the reviewer for raising these important points regarding fairness and visualization. To clarify, all hyperparameters and implementations follow the author-recommended defaults from the original publications.
>
> __First__, for Table 3, we follow the settings described in Sec. 3.1 and compare the performance-efficiency trade-offs between NICER and PANTHER in the Condensation setting. PANTHER is a GMM-based method controlled by a predefined number of mixture components $K$, while NICER is controlled by the maximum capacity $M$. By varying the capacities of both methods, we show that NICER achieves more stable performance across different settings and offers a better trade-off than PANTHER.
>
> __Second__, in Figure 8, as the reviewer notes, t-SNE can introduce artificial fragmentation and should not be overinterpreted. We should treat those as qualitative demonstrations which need to be coupled with quantitative metrics such as downstream predictive tasks or other qualitative results. To support this, we instead compute the average pairwise distance in the prototype original space for the first figure in Figure 8. __The pairwise distance is: PANTHER = 28.7399, NICER=40.12854__, indicating NICER’s concepts are more diverse. With this result, we can demonstrate using both tSNE and UMAP or other visualization tools showing the same conclusion, as shown in the UMAP version in this link: [UMAP for Fig 8](https://github.com/anon-user-0/NICER-submission/blob/main/analysis_results/UMAP_NICER/umap_fig8.jpg).
>
> __Q4. "The way the paradigm is theoretically framed reminds about sparse coding with overcomplete dictionaries: Aharon, M., Elad, M., & Bruckstein, A. (2006). K-SVD: An algorithm for designing overcomplete dictionaries for sparse representation. IEEE Transactions on signal processing, 54(11), 4311-4322. I would like that authors try to analize analogies and differences."__
>
> We thank the reviewer for bringing this work to our attention. While NICER and K-SVD both employ iterative optimization and aim to achieve compression, there are two fundamental differences.
>
> __First__, K-SVD learns a dictionary $D$ with a fixed number of atoms $K$, where each atom $d_k$ is updated via SVD. In contrast, NICER jointly learns both the concept prototypes and their effective capacity in a fully data-driven, nonparametric manner. K-SVD cannot efficiently achieve this: choosing a large fixed $K$ introduces unnecessary redundancy, while searching for an appropriate $K$ for each WSI negates the purpose of reducing computational cost and aggravates the original redundancy problem that NICER is designed to solve.
>
> __Second__, K-SVD assumes a linear reconstruction model with a predefined number of dictionary atoms, which limits its flexibility for modeling the highly heterogeneous and nonlinear feature distributions found in gigapixel WSIs. In our setting, the goal is to condense data for arbitrary downstream tasks, and a fixed-capacity linear dictionary can lead to suboptimal prototype quality and reduced predictive performance. NICER’s nonparametric formulation and parameterization avoids these constraints by adapting its capacity to each slide’s complexity, enabling more expressive and robust condensation.

---

> > ### Comment · Reviewer_SG2S · 2025-11-27
> > **Q3 and Q4**
> >
> > The UMAP provided for figure 8 does not appear convincing. Why a significant number of prototypes are outside the embedding? Especially for PANTHER? Also why not showing how those prototype look, like showcased in original PANTHER paper?
> > The answer to Q4 is convincing; authors can add some of this comparison in the manuscript.

---

> ### Author Response · Authors · 2025-11-24
>
> __Q5. "Scalability performance: what do you expect as trade-off for condensation? How much does it cost to perform condensation instead of keeping the full WholeOfBag?"__
>
> We thank the reviewer for the thoughtful comment. The table below summarizes the computation costs for NICER and PANTHER across different stages. As shown, NICER’s main overhead comes from the condensation step; however, this step is performed __only once per WSI__ when it is added to the database and can then be reused for all downstream training and testing.
>
> Importantly, NICER achieves a __34× reduction in total storage__ while maintaining comparable - or even superior-performance relative to using the full Whole Bag. This demonstrates that NICER offers strong practical benefits with minimal additional cost.
>
> | Computation Metrics                 | Whole Bag    | PANTHER      | NICER        |
> |-------------------------------------|--------------|--------------|--------------|
> | Total Storage (GB)                  | 194          | 3.2          | 5.7          |
> | Condensation GPU per WSI (MB)       | -            | 5.30         | 9.17         |
> | Condensation time per WSI (s)       | -            | 1.5103       | 0.044        |
> | Training time per condensed WSI (s) | 0.05361      | 0.00659      | 0.00449      |
> | Inference time (Condensation) (s)   | 0.00655      | 0.00655      | 0.00655      |
> | Inference time (Prototyping) (s)    | 0.00655      | 0.00035      | 0.00108      |
> | Kappa                               | 87.17 ± 0.90 | 81.41 ± 2.38 | 88.47 ± 1.56 |
> | Bal. Acc.                           | 93.57 ± 0.41 | 90.72 ± 1.14 | 94.28 ± 0.76 |
> | F1                                  | 93.58 ± 0.45 | 90.69 ± 1.21 | 94.23 ± 0.79 |
>
> __Q6. Manuscript Writing Revision__. Thank you for pointing these out. We will revise these to improve the manuscript.
>
> ---
>
> Given the above, we hope the Reviewer can take another look and consider upgrading the rating if our responses have addressed all concerns sufficiently. Otherwise, please let us know if you still have questions for us

---

> > ### Comment · Reviewer_SG2S · 2025-11-27
> > **Answer to Q5 and Q6**
> >
> > The answer to Q5 partially addresses the raised concern, and showcases slight quantitative improvements wrt PANTHER. Still a direct comparison of the quality of condensed prototypes is required.

---

> ### Author Response · Authors · 2025-11-28
>
> We thank the reviewer for constructive feedback and follow-up questions.
>
> __Q1. "Did you already update the table in the manuscript? I cannot find clear indication on what is the final number of concepts, and the answer to my question does not appear clear. Please provide also preliminary results as official comment. Also for Q2, I would expect that Fig. 8 is redone with UMAP, and for reference, please consider also PCA, or justify why to discard it."__
>
> As requested by the reviewer, we report the average number of prototypes produced by NICER. The results show that for larger and more complex WSIs (e.g., NSCLC, LUAD, BRCA), NICER naturally allocates more capacity to preserve the underlying conceptual diversity in the slides. We have added this result and discussion in our revised manuscript (Table 5, line 1000).
>
> | PANDA | NSCLC  | LUAD (Survival) | BRCA (Survival) |
> |-------|--------|-----------------|-----------------|
> | 62.80 | 154.67 | 157.40          | 158.60          |
>
> __Q2. "The UMAP provided for figure 8 does not appear convincing. Why a significant number of prototypes are outside the embedding? Especially for PANTHER? Also why not showing how those prototype look, like showcased in original PANTHER paper?"__
>
> We thank the reviewer for bringing this to our attention. We would like to emphasize two points:
> (1) prototype markers falling outside the embedded region are indeed an artifact of 2D projection methods, and (2) the contrast between NICER’s well-spread prototypes and PANTHER’s tightly clustered ones appears consistently across multiple projection techniques, indicating that this behavior reflects the underlying methods rather than projection noise.
>
> To clarify this distinction, we provide visualizations using several different projection methods below.
>
> https://github.com/anon-user-0/NICER-respons-1/blob/main/all_projections.jpg
>
> __UMAP Artifacts__. Both PANTHER and NICER produce latent prototypes that summarize the original patch features. Because these prototypes reflect blended information, some are not strongly tied to any specific patch neighborhood, leading UMAP to push them toward the periphery of the embedding. For PANTHER in particular, many prototypes lie in a tight region of the high-dimensional space and share similar, weak connections to the patch manifold (which is consistent across 2D projection methods). UMAP therefore places them together - and often further from the dense patch cloud - explaining why several PANTHER prototypes appear outside the main embedded region. This observation is not shown in PCA and t-SNE projections, indicating that it is an artifact of UMAP.
>
> __Consistent diversity of NICER across 2D projection methods.__ By examining multiple standard projection methods, we observe a consistent pattern: the prototypes learned by PANTHER occupy a very limited region relative to the original feature distribution. This consistency across visualizations suggests that the lack of expressiveness is not a projection artifact, but rather reflects a genuine property of PANTHER’s prototypes in the high-dimensional space. We then explain this behavior as follow,
>
> __First__, the observed behavior of PANTHER is caused by its underlying GMM formulation. PANTHER effectively assumes that the WSI feature bag can be modeled as a set of well-separated Gaussian clusters and attempts to fit mixture components accordingly. However, WSI features are highly heterogeneous: capturing cellular morphology, tissue architecture, cancer subtypes, and other complex patterns, which rarely conform to such linear Gaussian structures. As a result, the mixture components tend to collapse toward one another in the high-dimensional space, producing prototype centers that are far less dispersed than the original feature patches. This collapse naturally persists across all projection methods.
>
> __Second__, this interpretation is also supported by our patch - concept assignment visualizations (image: https://github.com/anon-user-0/NICER-submission/blob/main/analysis_results/Biology_explain/bio.jpg ). PANTHER’s assigned regions appear noisier and less coherent than those produced by NICER, suggesting that some prototypes may not correspond to meaningful morphological patterns and may introduce false positives.
>
> __In contrast__, NICER is explicitly designed to preserve the diversity of the original feature bag while still achieving strong compression. By formulating pattern exploration as a retrieval problem, NICER avoids the linear assumptions inherent to GMMs. Moreover, its iterative optimization enables the condensed concepts to continually revisit and refine their representations against the original observations. These design choices help NICER maintain richer and more faithful diversity, which is consistently reflected across all projection methods.

---

> > ### Author Response · Authors · 2025-11-28
> >
> > __Q3. "The answer to Q5 partially addresses the raised concern, and showcases slight quantitative improvements wrt PANTHER. Still a direct comparison of the quality of condensed prototypes is required."__
> >
> > To address the reviewer’s concerns, we visualize example patches from each concept prototype learned by PANTHER.
> >
> > https://github.com/anon-user-0/NICER-respons-1/blob/main/patch_examles.jpg
> >
> > From these figures, we can see that NICER indeed learns tissue with different morphological and visual information, showing that NICER can capture biological information. This also aligns with our patch-prototype assignment map: https://github.com/anon-user-0/NICER-submission/blob/main/analysis_results/Biology_explain/bio.jpg
> >
> > However, we agree that assigning definitive biological semantics to each prototype requires expert pathological review. While this is beyond the scope of the current paper, we view it as a valuable direction for future collaboration with certified pathologists based on the preliminary results. For this work, we focus on demonstrating NICER’s novel formulation, efficiency benefits, and strong empirical performance, which hold independently of detailed biological annotation.
> >
> > ---
> >
> > Please let us know if you have any questions for us.

---

### Official Review · Reviewer_sdeQ · 2025-10-30

**Soundness:** 2
**Presentation:** 1
**Contribution:** 2
**Rating:** 2
**Confidence:** 3

**Summary:**

The paper introduces NICER, a nonparametric unsupervised data condensation framework for whole-slide histology images (WSIs). NICER adaptively determines the number of prototypes per slide by formulating condensation as a hierarchical probabilistic model (features -> patterns -> concepts). The goal is to balance information preservation and computational efficiency. The method is evaluated across four datasets and on two main tasks: cancer subtyping and survival prediction. The authors report consistent improvements over prior works in the field of prototype learning, such as PANTHER and ProtoCount.

**Strengths:**

1. Relevant problem: WSI condensation is a well-known bottleneck in computational pathology, as slides are composed of a very large number of tiles, many of which are redundant. Tackling this problem in an unsupervised way is of practical and methodological interest.
2. Comprehensive empirical evaluation: The experiments cover multiple datasets and tasks with consistent baselines and metrics. The reported gains over comparable prototype-based approaches are systematic.

**Weaknesses:**

1. Methodological Ambiguities
Despite the heavy probabilistic framing, the method is mathematically under-specified:
- The hierarchical model 𝑃(𝐻,𝑍∣Ω)  is introduced but not clearly derived or grounded in an actual probabilistic generative process.
- Many equations (e.g., Eq. 2–9) appear to be reformulations of clustering or assignment heuristics, rather than principled probabilistic inference steps.
- The “nonparametric” property arises mainly from pruning unused prototypes, not from a Bayesian nonparametric process.
- Numerous notation and formulation issues undermine clarity:
  * 𝒩  (Gaussian) is confused with ℕ  (natural numbers).
  * Confusion between probability laws and densities in expressions such as log(𝒩(⋅)).
  * Inconsistencies between Eq. 6 and Eq. 8
  * Typos (max vs. min, line 173).
  * Equation 4 is ill-defined.
  * Some variables are not defined (e.g. d l173, α l229).
  * “Since both hi and z∗ (i) are ℓ2-normalized embeddings, we approximate ∥hi∥2 ≈ ∥z∗ (i)∥2 ≈ 1”.  It is equal to 1, no approximation here.
- Overall, the pattern exploration method relies mostly on deterministic clustering mechanisms, with the Bayesian formalism serving mainly as an interpretive framework rather than a fully realized inference model.

2. Performance Claims
- The reported “up to 90% performance gains” (abstract) are misleading.
- Table 1 shows that NICER features still underperform the vanilla method (whole bag). Ideally, condensation should reduce redundancy while maintaining or improving performance, as demonstrated in PANTHER paper; here, the gain comes at the cost of lower performance.
- The paper should include comparisons to other whole-slide representation models (e.g., GigaPath, GigaSSL, PRISM, HIPT).

3. Biological Interpretability
- The biological or histological meaning of prototypes is under-explored:
  * What morphological patterns do prototypes capture?
  * Visualization of tile-prototype associations or spatial prototype maps would significantly strengthen the claim of the paper.

**Questions:**

- Some mathematical expressions are difficult to follow because several variables are not defined when first used. I recommend clearly defining all variables and revising the math sections to make them more precise, accurate, and rigorous.
- Simplify the mathematical formulations to highlight the core mechanism.
- What morphological or histological patterns do the learned prototypes capture?
- Could the authors provide visualizations of tile–prototype associations or spatial prototype maps to support biological interpretability?

---

> ### Author Response · Authors · 2025-11-24
>
> We thank the reviewer for spending their time on reviewing our work.
>
> ## Method Clarification.
>
> __Q1."Overall, the pattern exploration method relies mostly on deterministic clustering mechanisms, with the Bayesian formalism serving mainly as an interpretive framework rather than a fully realized inference model."__
>
> __NICER is a MAP Inference Model.__ We thank the reviewer for raising this point, and we agree that NICER corresponds to a MAP solution of a Bayesian nonparametric process rather than a full posterior inference. In general, Bayesian nonparametric models include a finite set of learnable parameters together with a data-dependent, potentially unbounded collection of latent variables that can be marginalized or estimated via MLE/MAP. NICER follows this formulation by using a random Poisson point process with a neuralized base measure to govern concept capacity.
>
> Importantly, we do not view the use of MAP inference as a limitation. A fully Bayesian treatment would marginalize over all latent variables and assignments, which would eliminate the identifiable prototypes required for WSI condensation and interpretability. In contrast, NICER deliberately adopts a MAP inference to preserve explicit concepts and assignments, making the resulting representation both interpretable and effective. We believe this should be viewed as a __novel and purposeful perspective on Bayesian nonparametric modeling__, tailored specifically to the goals of WSI condensation.
>
> __Q2. Mathematical notation confusion and declaration__
>
> We thank the reviewer for pointing this out. While these notations are widely used and generally understood in the machine learning literature, we agree that additional clarity can improve readability. As suggested, we __have revised the reviewer’s main concerns__ and provide brief explanations to ensure the presentation is self-contained, as shown in the rebuttal version. We will also revise all stylistic notation as suggested to improve clarity. We would like to clarify our mathematical framework as follows,
>
> __Nonparametric nature of NICER is not heuristic.__ The removal of unused concepts, which the reviewer refers to as heuristic pruning, is not heuristic even though the intended laymen narrative might have created such impression in exchange for better readability. It is in fact derived from a principled inference of the most probable mixture which is equivalently cast as maximizing a maxima search problem between global concepts and local patterns. The optimized match reveals in principle which concepts are not matched by any local patterns and should be logically considered outdated/unused. This whole process is reflected in Eqs (9).
>
> __Specified Probabilistic Model.__ The manuscript provides a complete probabilistic generative process under Bayesian prior and each equation follows directly from this. In particular, Sections 2.2 - 2.3 describe a full generative process: (1) A set of concept prototypes $\Omega = $ {$\omega_k$} is assumed, then (2) each pattern $z_m$ is drawn from a Gaussian distribution parameterized by exactly one concept via one-hot latent variable $b_m$ and given $\Omega$, under categorical prior (Eq. 4 - 5). Finally, each observed patch feature $h_i$ is generated from a Gaussian likelihood whose mean and covariance are determined by its relevant patterns (Eq. 2-3). This defines all random variables, distributions and conditional dependencies, leading to our complete probabilistic specification of $P(H, Z, b | \Omega)$. However, we agree that the generative story can be stated more clearly, and will add explicit sampling steps in our revision.
>
> __Eqs. (2-9) follow the model, not heuristics.__ Eq (2) is derived directly from the Gaussian assumption and the simplification to cosine similarity comes from standard expansion of the Gaussian log-likelihood when embeddings are $\ell_2$-normalized. Eqs. (4-7) introduces Bayesian prior for random variables in our generative process which come from their nature and requirements given the condensation problems. Lastly, Eq. 8 is the joint log-likelihood of $(Z, b)$ under the factorization of Eq. (6), and Eq. (9) results from applying the one-hot constraint on $b_m$, which is required since we are condensing the patterns into concepts. In short, we would like to emphasize that none of these equations are heuristic; they are mathematically required steps for MAP estimation in our model we define.

---

> ### Author Response · Authors · 2025-11-24
>
> ## Performance clarification
>
> __Q3. "Table 1 shows that NICER features still underperform the vanilla method (whole bag). Ideally, condensation should reduce redundancy while maintaining or improving performance, as demonstrated in PANTHER paper; here, the gain comes at the cost of lower performance."__
>
> We appreciate the reviewer for this concern. However, in our experiments, we should treat the Whole-Bag baseline as an upper-bound for all methods to approach, since this preserves all information contained in WSIs. In Table 1, we report the results in the condensation task, where MIL decoders are trained on a condensed dataset and evaluated on the original bag of features. This possibly makes the performance drop due to differences in training and testing data domains, with an advantage of computation and storage cost reduction.
>
> As can be seen from Table 1, PANTHER falls short of approaching this upper-bound while NICER comes much closer while reducing the cost significantly. This emphasizes how well NICER can solve the problem of computation in WSI research, as shown in a table below.
>
> | Computation Metrics                 | Whole Bag    | PANTHER      | NICER        |
> |-------------------------------------|--------------|--------------|--------------|
> | Total Storage (GB)                  | 194          | 3.2          | 5.7          |
> | Condensation GPU per WSI (MB)       | -            | 5.30         | 9.17         |
> | Condensation time per WSI (s)       | -            | 1.5103       | 0.044        |
> | Training time per condensed WSI (s) | 0.05361      | 0.00659      | 0.00449      |
> | Inference time (Condensation) (s)   | 0.00655      | 0.00655      | 0.00655      |
> | Inference time (Prototyping) (s)    | 0.00655      | 0.00035      | 0.00108      |
> | Kappa                               | 87.17 ± 0.90 | 81.41 ± 2.38 | 88.47 ± 1.56 |
> | Bal. Acc.                           | 93.57 ± 0.41 | 90.72 ± 1.14 | 94.28 ± 0.76 |
> | F1                                  | 93.58 ± 0.45 | 90.69 ± 1.21 | 94.23 ± 0.79 |
>
> __Q4. "The paper should include comparisons to other whole-slide representation models (e.g., GigaPath, GigaSSL, PRISM, HIPT)."__
>
> Our work targets a different problem setting from the methods suggested by the reviewer. This is why we did not include them as direct baselines. To elaborate, unlike previous work that aims to pre-train new foundation encoders for WSIs, both PANTHER and NICER build data-condensation techniques on top of existing pre-trained encoders. In NICER, the main contribution is a nonparametric unsupervised condensation framework that operates on top of these encoders and is orthogonal to developing them. Thus, rather than comparing NICER with previously pre-trained WSI encoders, the appropriate comparison is between NICER's and PANTHER's data condensation effectiveness under the same pre-trained encoder, where PANTHER has been shown to reach the best performance. This is how we organize our experiment.

---

> ### Author Response · Authors · 2025-11-24
>
> ## Biological Interpretation
>
> __Q5. "The biological or histological meaning of prototypes is under-explored: What morphological patterns do prototypes capture? Visualization of tile-prototype associations or spatial prototype maps would significantly strengthen the claim of the paper."__
>
> We thank the reviewer for raising this point. Following PANTHER’s evaluation protocol, we visualize the patch-concept assignments and overlay them on the original WSIs (see this link: [biology interpretation](https://github.com/anon-user-0/NICER-submission/blob/main/analysis_results/Biology_explain/bio.jpg) ). These visualizations show that NICER produces __clearer and more coherent region separation__, indicating that NICER groups morphologically similar patches more faithfully. This aligns with established observations in the WSI literature that similar tissue structures tend to cluster spatially [1].
>
> However, we agree that assigning definitive biological semantics to each prototype requires expert pathological review. While this is beyond the scope of the current paper, we view it as a valuable direction for future collaboration with certified pathologists. For this work, we focus on demonstrating NICER’s __novel formulation, efficiency benefits, and strong empirical performance__, which hold independently of detailed biological annotation.
>
> [1] Ho et al., Deep Learning-Based Objective and Reproducible Osteosarcoma Chemotherapy Response Assessment and Outcome Prediction. Am J Pathol. 2023
>
> ---
>
> Given the above, we hope the reviewer can take another look and consider upgrading the rating if our responses have addressed all concerns sufficiently. Otherwise, please let us know if you still have questions for us

---

### Official Review · Reviewer_9Nuz · 2025-11-03

**Soundness:** 2
**Presentation:** 2
**Contribution:** 2
**Rating:** 2
**Confidence:** 4

**Summary:**

The paper introduces NICER, a probabilistic nonparametric method for unsupervised data condensation of gigapixel histology slides. Unlike prior approaches that use a fixed number of prototypes per slide, NICER adapts the number of prototypes to each slide’s morphological complexity. It models prototype construction hierarchically, preserving diverse local patterns while enforcing compact global representation. Across multiple datasets, NICER achieves strong gains in F1 and efficiency over existing condensation and MIL baselines.

**Strengths:**

**Clear motivation and problem relevance.**
  The paper identifies an important challenge in computational pathology about how to compress extremely large WSIs while retaining morphological diversity.

**Readable structure and visuals.**
  Figures are generally clear and intuitive. The narrative follows a standard and familiar structure, making the paper easy to navigate.

**Potential applicability.**
  Adaptive prototype condensation could, in principle, generalize to other domains that require scalable representation learning, even though this is not empirically verified.

**Weaknesses:**

**Inflated novelty.**
The core mechanism of NICER is largely a reformulation of existing prototype learning and clustering frameworks such as PANTHER, OT-based embeddings, and Gaussian mixture condensation. The “pattern–concept” hierarchy is functionally equivalent to a coarse-to-fine clustering scheme, and the probabilistic factorization adds notation without introducing new learning objectives or inference principles. Overall, the contribution is incremental rather than conceptually novel.

**Inaccurate “nonparametric” claim.**
Although repeatedly described as nonparametric, NICER is a fully parametric neural framework with learnable embeddings and trainable parameters. Its capacity is bounded by predefined hyperparameters and adjusted via heuristic pruning. The approach does not perform genuine nonparametric inference or kernel-based adaptation. Hence, its “nonparametric” behavior is empirical and heuristic rather than statistical or data-driven.

**Unverifiable and internally inconsistent experimental comparisons.**
The paper does not clarify which configuration underlies the reported “SOTA” results, making them empirically unverifiable and potentially biased in favor of NICER. In addition, both PANTHER (and possibly other prototype-based methods) and the survival prediction tasks are sensitive to hyperparameters and data splits, which makes the reported improvements difficult to reproduce.

**Unsupported efficiency and generalization claims.**
The paper emphasizes a superior efficiency–performance trade-off and even a new paradigm for histological representation learning, yet provides no runtime, GPU memory, or throughput measurements. The reported metric reflects only compression ratio, not computational cost. Moreover, all experiments are confined to a bag-of-features assumption. This narrow scope and lack of computational evidence undermine both the efficiency and generalization claims.

**Questions:**

- Including additional baseline comparisons beyond PANTHER and OT-based methods If possible.

- If possible, provide a runtime and preprocessing time analysis. The preprocessing stages (patch extraction, embedding, and condensation) appear to dominate total computation time.  It would be helpful to quantify how much time each stage requires compared with end-to-end MIL training or inference.  This could clarify whether NICER improves overall pipeline efficiency.

---

> ### Author Response · Authors · 2025-11-24
>
> We thank the reviewer for their time on reviewing our work.
>
> __Q1. "The core mechanism of NICER is largely a reformulation of existing prototype learning and clustering frameworks such as PANTHER, OT-based embeddings, and Gaussian mixture condensation. The “pattern–concept” hierarchy is functionally equivalent to a coarse-to-fine clustering scheme, and the probabilistic factorization adds notation without introducing new learning objectives or inference principles. Overall, the contribution is incremental rather than conceptually novel."__
>
> We respectfully disagree that NICER is incremental or merely a reformulation of prior prototype-learning or clustering methods.
>
> __First__, NICER is, to our knowledge, __the first method to address whole-slide data distillation in a fully unsupervised and nonparametric manner__. Prior prototype-based approaches (e.g., PANTHER, OT embeddings, GMM variants) all assume a fixed prototype capacity and do not attempt to automatically adapt the number of prototypes to slide complexity. NICER directly solves this missing problem by casting prototype construction as a __probabilistic data condensation task__ that discovers both the appropriate number and structure of prototypes *without supervision*, a capability that does not exist in previous WSI or prototype-learning frameworks.
>
> __Second__, reducing NICER to clustering oversimplifies our proposed framework. NICER introduces a __two-level latent hierarchy - patterns and concepts - that interact following a generative model__. Patterns preserve diverse morphological signals and prevent premature collapse. Concepts then condense these patterns through probabilistic assignments with learnable categorical priors and Gaussian generators. Their __joint inference procedure__ - alternating optimization over continuous parameters and discrete one-hot latent variables - enables NICER to define unused concepts in a genuinely *nonparametric, data-driven way*.
>
> No prior clustering, GMM, or prototype-learning method uses this __division of roles__ (preservation to condensation) or this __hierarchical probabilistic coupling__. Thus, calling NICER “just clustering” misses the core novelty: the *interaction* between these two levels is what enables nonparametric and flexible condensation.
>
> __Last__, we emphasize that adopting prior components in a new framework should not be used as an indication of incrementality. Otherwise, going by this logic, generative pretraining is incremental because it is entirely based on the transformer decoder architecture. This is clearly not true. As such, we hope the reviewer will reconsider this.
>
> __Q2. Although repeatedly described as nonparametric, NICER is a fully parametric neural framework with learnable embeddings and trainable parameters. Its capacity is bounded by predefined hyperparameters and adjusted via heuristic pruning. The approach does not perform genuine nonparametric inference or kernel-based adaptation. Hence, its “nonparametric” behavior is empirical and heuristic rather than statistical or data-driven.__
>
> It appears that the reviewer might be referring to a specific class of non-probabilistic, non-parametric methods while our method is probabilistic non-parametric or more precisely, Bayesian non-parametric. In both cases, a model is categorized as “nonparametric” when its modeling capacity can grow with the complexity of the dataset but depending on the specific case, it might or might not have an additional set of learnable hyper-parameters.
>
> A Bayesian non-parametric model might have a finite number of learnable parameters in addition to a growing set of latent parameters whose size is data-driven and need to be marginalized out or inferred via MLE/MAP techniques. For example, a random process (e.g., Poisson point process) characterizes a prior over a space of countably infinite mixture models (hence, non-parametric) and but also has a finitely parameterized base measure.
>
> Our method is in fact based on a random Poisson point process with a neuralized base measure. Its nonparametric nature arises from its ability to adapt its capacity to the complexity of the WSI, allowing the set of concepts to shrink as needed. This adaptability is not driven by hand-crafted rules, but emerges naturally from the __data-driven condensation process__, which is explicitly optimized by solving Eq. (9) in our manuscript.
>
> The removal of unused concepts, which the reviewer refers to as heuristic pruning, is not heuristic even though the intended laymen narrative might have created such impression in exchange for better readability. It is in fact derived from a principled inference of the most probable mixture which is equivalently cast as __a maxima search problem between global concepts and local patterns__. The optimized match reveals in principle which concepts are not matched by any local patterns and should be logically considered outdated/unused. This whole process is reflected in Eqs (9).

---

> ### Author Response · Authors · 2025-11-24
>
> __Q3. "The paper does not clarify which configuration underlies the reported “SOTA” results, making them empirically unverifiable and potentially biased in favor of NICER. In addition, both PANTHER (and possibly other prototype-based methods) and the survival prediction tasks are sensitive to hyperparameters and data splits, which makes the reported improvements difficult to reproduce."__
>
>
> We thank the reviewer for raising concerns about reproducibility and clarify that all configurations underlying the results are defined explicitly in the manuscript
>
> The dataset, evaluation metrics, evaluation settings and baseline implementations are described in Sec. 3.1 and Appendix C.3. To ensure fairness, all baselines use the same pretrained encoder, the same training/test set splits and same MIL predictors, so the comparison isolates only the condensation mechanism.
>
> We also agree that PANTHER and other baselines can be sensitive to hyperparameters. For this reason, we use author-recommended default settings for all baselines. For NICER, we conduct different ablation studies to investigate its sensitivity to hyperparameters and report standard deviation to verify robustness.
>
> Finally, we release parts of our source code in this link: [source code](https://github.com/anon-user-0/NICER-submission) for experiment demonstration. The full source code will be released upon acceptance.
>
> __Q4. The paper emphasizes a superior efficiency–performance trade-off and even a new paradigm for histological representation learning, yet provides no runtime, GPU memory, or throughput measurements. The reported metric reflects only compression ratio, not computational cost.__
>
> We thank the reviewer for the concern. The table below summarizes total storage and computation costs across the main stages of a standard WSI pipeline under different condensation methods.
>
>
> | Computation Metrics                 | Whole Bag    | PANTHER      | NICER        |
> |-------------------------------------|--------------|--------------|--------------|
> | Total Storage (GB)                  | 194          | 3.2          | 5.7          |
> | Condensation GPU per WSI (MB)       | -            | 5.30         | 9.17         |
> | Condensation time per WSI (s)       | -            | 1.5103       | 0.044        |
> | Training time per condensed WSI (s) | 0.05361      | 0.00659      | 0.00449      |
> | Inference time (Condensation) (s)   | 0.00655      | 0.00655      | 0.00655      |
> | Inference time (Prototyping) (s)    | 0.00655      | 0.00035      | 0.00108      |
> | Kappa                               | 87.17 ± 0.90 | 81.41 ± 2.38 | 88.47 ± 1.56 |
> | Bal. Acc.                           | 93.57 ± 0.41 | 90.72 ± 1.14 | 94.28 ± 0.76 |
> | F1                                  | 93.58 ± 0.45 | 90.69 ± 1.21 | 94.23 ± 0.79 |
>
> While PANTHER and similar baselines are lightweight, NICER introduces only a small additional cost due to jointly solving for patterns, concepts, and their interactions. The main overhead comes from condensation, but this step is __performed once per WSI__ when it is added to the database. As a result, __inference time remains comparable to PANTHER__, while NICER delivers substantially higher downstream performance.
>
> Moreover, NICER reduces storage by 34× compared to keeping all WSI feature bags, improves end-to-end preprocessing, and achieves equal or superior accuracy. Overall, these results demonstrate NICER’s strong performance-computation trade-off.

---

> ### Author Response · Authors · 2025-11-24
>
> __Q5. "Moreover, all experiments are confined to a bag-of-features assumption. This narrow scope and lack of computational evidence undermine both the efficiency and generalization claims."__
>
> We thank the reviewer for raising this point, but we believe this assumption does not affect our contributions in WSI research. __This assumption is not a limitation unique to our work__ - it is the standard and dominant paradigm in computational pathology. All of our baselines and MIL predictors also operate under the same bag-of-features setting. This assumption is widely adopted because WSIs are gigapixel-scale and patch-based feature extraction with MIL setting is a computationally feasible pipeline used in practice.
>
> Given this shared foundation, our efficiency and generalization claims remain fully valid: NICER is compared fairly and directly against strong baselines under the exact same input representation. Our focus is on improving __condensation and representation quality on top of standard WSI features__, and the results consistently show that NICER yields superior performance and substantial storage/computation gains relative to all competing methods using the same bag-level inputs.
>
> While exploring spatial models is a promising future direction, evaluating NICER within the widely accepted bag-of-features framework is both appropriate and aligned with current WSI methodology.
>
> ---
>
> Given the above, we hope the Reviewer can take another look and consider upgrading the rating if our responses have addressed all concerns sufficiently. Otherwise, please let us know if you still have questions for us

---

### Official Review · Reviewer_FYhf · 2025-11-06

**Soundness:** 2
**Presentation:** 2
**Contribution:** 2
**Rating:** 2
**Confidence:** 4

**Summary:**

This paper introduces NICER, a framework for unsupervised data condensation of gigapixel whole-slide images (WSIs). The method aims to address the limitation of prior works that use a fixed number of prototypes, proposing instead a two-stage probabilistic model that adapts the prototype capacity to the complexity of each slide. It first extracts a large set of feature patterns to preserve information, then condenses them into a smaller set of concept prototypes. The authors claim this nonparametric approach achieves state-of-the-art performance on four histological datasets.

**Strengths:**

The paper identifies an important and relevant problem in computational pathology: the inadequacy of fixed-capacity representations for WSIs of varying complexity. The proposed two-stage approach of preservation followed by condensation is, at a conceptual level, an intuitive and promising direction.

**Weaknesses:**

1. The paper's central conceptual pillar is its "nonparametric" nature. However, the method is simply an iterative algorithm with a heuristic pruning step. This is a severe misrepresentation of the methodology.
2. The paper's entire premise is built on improving the trade-off between accuracy and efficiency. Yet, there is a complete and inexplicable absence of any empirical data regarding efficiency—no training times, no inference speeds, no memory usage comparisons.
tion.
3. The methodology is described at such a high level that it is impossible to reproduce. Key design choices (initialization, network architectures, reconciliation of top-k vs. top-1) are omitted.

**Questions:**

1. Can you provide a rigorous justification for using the term "nonparametric"? If not, are you willing to retract this claim and re-frame your contribution more accurately as an "adaptive-capacity" model?

2. Please provide a new table comparing NICER against key baselines (e.g., PANTHER, H2T) on wall-clock training and inference time per WSI, as well as peak GPU memory consumption.

3. Please provide the exact implementation details necessary for reproducibility: (a) How was the initial pattern set Z generated? (b) What are the specific architectures and hyperparameter settings? (c) Please clarify the discrepancy between the "top-k" description and the "top-1" formulation in Eq. 2 and state precisely what objective was implemented.

---

> ### Author Response · Authors · 2025-11-24
>
> We thank the reviewer for spending their time on reviewing our work.
>
> __Q1. "The paper's central conceptual pillar is its "nonparametric" nature. However, the method is simply an iterative algorithm with a heuristic pruning step. This is a severe misrepresentation of the methodology."__
>
> We respectfully disagree with the characterization that our method is “simply an iterative algorithm with a heuristic pruning step,” as this oversimplifies the core methodology. The impression likely stems from our choice to present the method in an accessible, high-level narrative. On the technical level, all components are mathematically coupled through a __principled, neuralized random point process__, and the pruning behavior emerges naturally from this formulation rather than from ad-hoc heuristics.
>
> In particular, the nonparametric nature of our framework arises from its ability to adapt the capacity to the complexity of the data, allowing the set of concepts to grow or shrink as needed. This adaptability is not driven by hand-crafted rules, but emerges naturally from the data-driven condensation process, which is explicitly optimized by solving Eq. (9) in our manuscript. This process is learned via a principled MAP inference on a neuralized random point process, as shown in our generative model (Eq (1)) and optimization (Eq (9)).
>
> __The removal of unused concepts happens when a global concept is not matched to any local pattern__. This emerges directly from our principled inference procedure, with the implementation structured to maintain computational efficiency. For this reason, it should not be interpreted as a heuristic or ad-hoc modification of the underlying model. By the same reasoning, many nonparametric clustering algorithms, such as DP-means, would have to be labeled “heuristic,” since they also initialize or prune clusters based on criteria inherent to the model. Our approach follows this well-established principle: model capacity adjusts automatically in response to the data, consistent with the nonparametric paradigm.
>
> __Q2. Computation comparison (training, inference time and GPU)__
>
> We thank the reviewer for the concern. The table below summarizes total storage and computation costs across the main stages of a standard WSI pipeline under different condensation methods.
>
> | Computation Metrics                 | Whole Bag    | PANTHER      | NICER        |
> |-------------------------------------|--------------|--------------|--------------|
> | Total Storage (GB)                  | 194          | 3.2          | 5.7          |
> | Condensation GPU per WSI (MB)       | -            | 5.30         | 9.17         |
> | Condensation time per WSI (s)       | -            | 1.5103       | 0.044        |
> | Training time per condensed WSI (s) | 0.05361      | 0.00659      | 0.00449      |
> | Inference time (Condensation) (s)   | 0.00655      | 0.00655      | 0.00655      |
> | Inference time (Prototyping) (s)    | 0.00655      | 0.00035      | 0.00108      |
> | Kappa                               | 87.17 ± 0.90 | 81.41 ± 2.38 | 88.47 ± 1.56 |
> | Bal. Acc.                           | 93.57 ± 0.41 | 90.72 ± 1.14 | 94.28 ± 0.76 |
> | F1                                  | 93.58 ± 0.45 | 90.69 ± 1.21 | 94.23 ± 0.79 |
>
> While PANTHER and similar baselines are lightweight, NICER introduces only a small additional cost due to jointly solving for patterns, concepts, and their interactions. The main overhead comes from condensation, but this step is performed __once per WSI__ when it is added to the database. As a result, __inference time remains comparable to PANTHER__, while NICER delivers substantially higher downstream performance.
>
> Moreover, NICER reduces storage by __34×__ compared to keeping all WSI feature bags, improves end-to-end preprocessing, and achieves equal or superior accuracy. Overall, these results demonstrate NICER’s strong performance–computation trade-off.

---

> ### Author Response · Authors · 2025-11-24
>
> __Q3. Please provide the exact implementation details necessary for reproducibility: (a) How was the initial pattern set Z generated? (b) What are the specific architectures and hyperparameter settings? (c) Please clarify the discrepancy between the "top-k" description and the "top-1" formulation in Eq. 2 and state precisely what objective was implemented.
> Thank you for bringing this to our attention. We would like to provide more details about the implementation details.__
>
> __Initialization of Z.__ The initial pattern set was first randomly generated as a sample from a multivariate Gaussian distribution. We follow the standard convention of initializing learnable parameters in deep learning literature.
>
> __Implementation details.__ Since all of MIL architectures in our experiments follow standard deep learning decoder with multiple layers, we follow their implementations with 1 layer to highlight the representation learned by our method. This design follows experimental settings of prior solutions (e.g., PANTHER) which use a MLP for experimenting. We would like to refer the reviewer to Appendix C.3. in our manuscript for more details on hyperparameter settings.
>
> __Top-$\kappa$ and Top-1.__ Top-$\kappa$ is a direct extension of the top-1 formulation in Eq. (2), which we reinterpret as a retrieval-based selection process (lines 180–181). In our implementation, the objective becomes cosine distance:
> $
> -\sum_{z_m \in Z(i)} \langle h_i, z_m \rangle,
> $
> where $Z(i)$ denotes the top-$\kappa$ patterns retrieved for patch $i$. This encourages each patch embedding to align, via cosine similarity, with its $\kappa$ most relevant patterns rather than only a single one, as in Eq.(2). This formulation follows standard retrieval-based learning [1] and naturally accommodates patches that contain multiple semantic patterns. Any overlap or redundancy among the retrieved patterns is subsequently resolved by the condensation process.
>
> [1] Wang et al., Learning to prompt in continual learning, CVPR 2023.
>
> ---
>
> Given the above, we hope the Reviewer can take another look and consider upgrading the rating if our responses have addressed all concerns sufficiently. Otherwise, please let us know if you still have questions for us

---

> ### Comment · Reviewer_FYhf · 2025-11-28
>
> After carefully reading the comments from other reviewers and the author's response, I believe that this paper does not currently meet the publication standards for the ICLR conference. Therefore, I still preserve the original score.

---

### Official Review · Reviewer_St93 · 2025-11-07

**Soundness:** 3
**Presentation:** 3
**Contribution:** 3
**Rating:** 6
**Confidence:** 5

**Summary:**

The paper introduces NICER, a nonparametric, unsupervised condensation framework for gigapixel WSIs. Instead of fixing a per-slide prototype budget, NICER first over-preserves morphology by learning slide-specific patterns (redundant by design), then condenses them into concept prototypes, pruning unused concepts so the final count adapts to slide complexity. The formulation is probabilistic with latent pattern–concept assignments optimized via alternating updates. Across four datasets (PANDA, NSCLC, BRCA, LUAD) and two tasks (subtyping, survival), NICER outperforms prior unsupervised prototype learners (DeepSets, ProtoCount, H2T, OT, InfiniteGPFA, PANTHER) and maintains a favorable accuracy–efficiency trade-off relative to Whole-Bag features (Tables 1–2, pp. 6–7; Table 3, p. 8). Ablations show how initial pattern count M and top-κ affect capacity and saturation (Fig. 4, p. 7; Table 3, p. 8).

**Strengths:**

Addresses a real bottleneck: avoids one-size-fits-all prototype budgets by adapting capacity per slide via a principled pruning mechanism (Sec. 2.3–2.4, pp. 4–5).

Balanced preservation→efficiency: the two-stage “intentional redundancy then condensation” reduces early information loss and retains rare morphology (Intro & Fig. 3, pp. 2–3).

Strong, broad empirical results: consistent wins over multiple unsupervised baselines (including the latest PANTHER approach) and across three MIL heads (ABMIL/DSMIL/ILRA) for subtyping and survival (Tables 1–2, pp. 6–7).

Trade-off evidence: clear performance–compression curves and sensitivity analyses (Table 3 & Fig. 4, pp. 8–7; Figs. 5–6, p. 7).

Practical framing: operates on foundation-encoder features, with implementation details and ablations that are actionable.

**Weaknesses:**

High Complexity and Computational Cost: NICER’s sophistication comes at the cost of increased complexity. The method involves an iterative two-stage learning process with many latent variables, which is more complicated to implement and tune than simpler one-step clustering or standard MIL models. The paper mentions using a 46GB GPU for experiments, indicating substantial memory and computation requirements. This could hinder adoption in practice – laboratories with limited computational resources might struggle to run NICER, whereas simpler methods (e.g. k-means or fixed GMM prototypes) are more lightweight. The authors do not report runtime comparisons, so the efficiency trade-off of NICER’s richer modeling is not fully clear.

Hyperparameter Sensitivity: While NICER is nonparametric in that it determines prototype counts automatically, it still requires setting certain hyperparameters that can affect results. In particular, the initial number of patterns (M) and the patch-to-pattern association limit (top-κ) must be chosen. The authors’ ablations show that too small an M can hurt performance and too large a κ can diffuse patch information. Thus, practitioners need to choose these carefully (the paper finds e.g. M≈200 and κ≈3 work well). NICER’s performance might degrade if these are mis-specified for a new dataset. In contrast, some simpler baselines have fewer tunable parameters. This points to a potential limitation in ease-of-use: despite its adaptive nature, NICER isn’t completely “hands-off” to configure.

Model Assumptions and Theoretical Guarantees: NICER relies on a probabilistic model (e.g. assuming patch features are approximately Gaussian around pattern means). These assumptions, while reasonable, are not deeply validated. Additionally, the optimization is heuristic (alternating updates for patterns, concepts, and assignments). The paper provides no formal proof of convergence or bounds on information loss during condensation. A skeptical reader might question whether the gains come from the sophisticated model or simply from clever engineering choices. Some discussion or theoretical insight into why the two-stage approach outperforms single-stage clustering (beyond empirical observation) would strengthen the work.


Novelty scope. The main novelty is the nonparametric per-slide capacity and the explicit pattern→concept condensation; however, the broader ideas (unsupervised prototyping, mixture-like modeling, hierarchical abstraction) overlap with existing lines. The paper could sharpen what is mathematically distinct from fixed-K GMM or from PANTHER’s soft assignment.

Clinical relevance and interpretability. While the paper demonstrates strong quantitative performance on subtyping and survival prediction, histological subtyping is a task pathologists routinely perform using visual cues. To establish clinical utility, it would be important to show that the concept prototypes learned by NICER correspond to morphologic patterns recognized by human experts (e.g., tumor regions, stroma, necrosis). The absence of such interpretability analysis limits the translational impact of the work.

External validity and batch effects. The evaluation relies exclusively on public TCGA and PANDA datasets, which are known to exhibit site-specific staining and preprocessing artifacts. These batch effects can inflate in-domain performance while limiting generalization. Validation on independent institutional or multi-center cohorts would provide stronger evidence that NICER’s adaptive condensation generalizes across data sources.

**Questions:**

Prototype Budget Determination: Could you clarify how NICER decides the final number of prototypes per slide? Is there an implicit threshold or stopping criterion in the generative model that prunes “redundant concepts”? For example, do you fix an initial maximum (M patterns or concepts) and then drop those with negligible assigned patches? Understanding what controls the adaptive prototype count (and how variable it is across slides) would help gauge the method’s robustness.

Rationale for Two-Stage Condensation: What is the key advantage of the hierarchical patterns→concepts approach versus a single-stage nonparametric clustering of patches? In principle one might try a Dirichlet Process or adaptive K-means on the patch embeddings directly. Does the two-step process (intentional redundancy then merging) simply preserve rare features better, or does it also aid optimization stability? Any insight or experiments comparing NICER’s two-stage pipeline to a one-stage variant would clarify why the hierarchy is crucial.

Computational Efficiency: How do the training time and memory usage of NICER compare to simpler prototype methods like PANTHER or to standard MIL models? The method appears resource-intensive; for instance, did you need to process one WSI at a time, or were multiple slides optimized in parallel? Any data on runtime per slide or per epoch would be appreciated. This information would help readers assess the practicality of NICER for large-scale or real-time applications.

Feature Extraction and End-to-End Learning: Are the patch features used in NICER fixed from a pretrained model, or were they trained/finetuned as part of this work? If fixed, did you observe any failure cases attributable to feature quality? And do you anticipate gains if one jointly learned the feature encoder with the NICER framework? It would be interesting to know if integrating representation learning (perhaps via a multi-task or self-supervised loss) was attempted, or if not, why the decoupled approach was chosen.

Position & Context: NICER currently treats the WSI as an unordered bag of patch features, focusing on morphological content. Might incorporating spatial context improve the prototypes (for example, ensuring that “concepts” correspond to contiguous regions or specific structures in tissue)? Some recent works add coordinate information or model the WSI as a graph to capture architecture. Did you consider extensions of NICER to encode spatial relationships between patches or to enforce that selected prototypes are spatially diverse? This could be relevant for tasks like tumor localization, so we wonder if it’s a plausible future direction.

Comparison with Other Unsupervised Methods: You included an “Infinite GPFA” baseline (Yu et al., 2025) which, like NICER, aims to learn latent factors without fixing their number. That method underperformed significantly. Can you shed light on why NICER achieves better results than InfiniteGPFA or other nonparametric clustering approaches? For example, is it due to NICER’s alternating optimization capturing more variance, or the specific way NICER handles patch-to-prototype assignments (top-κ redundancy, etc.)? A deeper explanation would highlight what design choices are most critical for NICER’s success relative to earlier approaches.

Clinical alignment of prototypes. Given that histologic subtyping is a task pathologists perform visually, have you examined whether the discovered prototypes correspond to interpretable histopathologic structures or features that pathologists recognize? For instance, do the top-activating patches for individual prototypes map to tumor, stroma, or inflammatory regions in a way that aligns with diagnostic reasoning? Such validation would help contextualize the model’s relevance to practical pathology workflows.

Cross-institutional validation. TCGA datasets contain known batch effects and pre-analytic heterogeneity. Have you tested NICER’s generalization when trained on one institution and tested on another (e.g., PANDA KRLS → RUMC split or external private cohorts)? If not, could the authors discuss how NICER’s adaptive, nonparametric mechanism might mitigate or exacerbate domain shifts across institutions?

---

> ### Author Response · Authors · 2025-11-24
>
> We thank the reviewing for the constructive comments.
>
> __Q1. Prototype Budget Determination Mechanism. "Could you clarify how NICER decides the final number of prototypes per slide? Is there an implicit threshold or stopping criterion in the generative model that prunes “redundant concepts”? For example, do you fix an initial maximum (M patterns or concepts) and then drop those with negligible assigned patches? Understanding what controls the adaptive prototype count (and how variable it is across slides) would help gauge the method’s robustness."__
>
> Thank you for this helpful suggestion. In NICER, the capacity of the concept set is controlled through the upper bound ($M$), and the nonparametric mechanism automatically determines the final number of concepts based on slide complexity. This design is important because WSIs are highly heterogeneous, making it difficult to predefine an appropriate target number of prototypes.
>
> As you noted, it is valuable to report the final number of concepts. In practice, we observe that the final concept count remains stable even when increasing the upper bound ($M$), as long as the slide complexity is fixed. We highlight this behavior, which emphasizes stable learning and robustness of NICER  in Table 3 and Figure 9 (with reported number of concepts), and will make this point more explicit in the revision.
>
> __Q2. Rationale for Two-Stage Condensation. "What is the key advantage of the hierarchical patterns→concepts approach versus a single-stage nonparametric clustering of patches? In principle one might try a Dirichlet Process or adaptive K-means on the patch embeddings directly. Does the two-step process (intentional redundancy then merging) simply preserve rare features better, or does it also aid optimization stability? Any insight or experiments comparing NICER’s two-stage pipeline to a one-stage variant would clarify why the hierarchy is crucial."__
>
> We thank the reviewer for highlighting this point. NICER outperforms other nonparametric clustering methods primarily due to (1) its separation of pattern exploration and concept condensation, and (2) its alternating optimization scheme.
>
> __First__, nonparametric Bayesian models apply a single-level factorization directly on all patch features. This often leads to over-clustering or noisy latent factors because every patch contributes equally to the global model. In contrast, NICER introduces a pattern exploration stage that selects a small set of informative pattern vectors using top-κ retrieval. This preserves meaningful morphological variation while filtering out noise and redundant patches before condensation, giving NICER a much cleaner set of inputs for concept estimation, as shown in the following comparison:
>
> | Method  | Decoder | Kappa         | Bal. Acc.     | F1            |
> |---------|---------|---------------|---------------|---------------|
> | KMeans  | ABMIL   | 84.57 ± 1.94  | 92.52 ± 1.53  | 92.59 ± 1.51  |
> | DPMeans |         | 83.98 ± 4.39  | 92.00 ± 2.09  | 91.96 ± 2.25  |
> | DPM     |         | 24.03 ± 4.18  | 62.00 ± 2.16  | 61.59 ± 2.51  |
> | NICER   |         | 89.73 ± 1.81  | 94.84 ± 0.92  | 94.87 ± 0.91  |
> | KMeans  | DSMIL   | 81.42 ± 1.10  | 90.73 ± 0.54  | 90.70 ± 0.56  |
> | DPMeans |         | 81.43 ± 1.12  | 90.78 ± 0.60  | 90.70 ± 0.56  |
> | DPM     |         | 16.26 ± 6.20  | 58.14 ± 3.06  | 56.96 ± 4.44  |
> | NICER   |         | 84.61 ± 1.56  | 92.30 ± 0.74  | 91.34 ± 0.79  |
> | KMeans  | ILRA    | 52.96 ± 16.00 | 76.64 ± 7.88  | 74.74 ± 9.56  |
> | DPMeans |         | 40.51 ± 27.49 | 70.44 ± 13.81 | 65.37 ± 19.55 |
> | DPM     |         | 8.64 ± 2.24   | 54.30 ± 1.08  | 54.00 ± 1.76  |
> | NICER   |         | 88.47 ± 1.56  | 94.28 ± 0.76  | 94.23 ± 0.79  |
>
> __Second__, NICER’s alternating optimization lets the model iteratively adjust both patterns and concepts in response to each other. If pattern selection is imperfect or condensation is too coarse, NICER can revisit the original features and refine assignments. Other methods’ posterior-based updates do not allow this direct feedback loop, so early clustering errors tend to propagate, often leading to suboptimal or overly fragmented latent factors.

---

> ### Author Response · Authors · 2025-11-24
>
> __Q3. Computation Efficiency. Computational Efficiency: How do the training time and memory usage of NICER compare to simpler prototype methods like PANTHER or to standard MIL models? The method appears resource-intensive; for instance, did you need to process one WSI at a time, or were multiple slides optimized in parallel? Any data on runtime per slide or per epoch would be appreciated. This information would help readers assess the practicality of NICER for large-scale or real-time applications.__
>
> We thank the reviewer for the concern. The table below summarizes total storage and computation costs across the main stages of a standard WSI pipeline under different condensation methods.
>
> | Computation Metrics                 | Whole Bag    | PANTHER      | NICER        |
> |-------------------------------------|--------------|--------------|--------------|
> | Total Storage (GB)                  | 194          | 3.2          | 5.7          |
> | Condensation GPU per WSI (MB)       | -            | 5.30         | 9.17         |
> | Condensation time per WSI (s)       | -            | 1.5103       | 0.044        |
> | Training time per condensed WSI (s) | 0.05361      | 0.00659      | 0.00449      |
> | Inference time (Condensation) (s)   | 0.00655      | 0.00655      | 0.00655      |
> | Inference time (Prototyping) (s)    | 0.00655      | 0.00035      | 0.00108      |
> | Kappa                               | 87.17 ± 0.90 | 81.41 ± 2.38 | 88.47 ± 1.56 |
> | Bal. Acc.                           | 93.57 ± 0.41 | 90.72 ± 1.14 | 94.28 ± 0.76 |
> | F1                                  | 93.58 ± 0.45 | 90.69 ± 1.21 | 94.23 ± 0.79 |
>
> While PANTHER and similar baselines are lightweight, NICER introduces only a small additional cost due to jointly solving for patterns, concepts, and their interactions. The main overhead comes from condensation, but this step is performed __once per WSI__ when it is added to the database. As a result, __inference time remains comparable to PANTHER__, while NICER delivers substantially higher downstream performance. The note of using a 48GB GPU is misleading because it is our maximum computation capacity and does not reflect real GPU assumed.
>
> Moreover, NICER reduces storage by __34×__ compared to keeping all WSI feature bags, improves end-to-end preprocessing, and achieves equal or superior accuracy. Overall, these results demonstrate NICER’s strong performance-computation trade-off.
>
> __Q4. Feature Extraction and End-to-end Learning. Feature Extraction and End-to-End Learning: Are the patch features used in NICER fixed from a pretrained model, or were they trained/finetuned as part of this work? If fixed, did you observe any failure cases attributable to feature quality? And do you anticipate gains if one jointly learned the feature encoder with the NICER framework? It would be interesting to know if integrating representation learning (perhaps via a multi-task or self-supervised loss) was attempted, or if not, why the decoupled approach was chosen.__
>
> We appreciate the reviewer’s thoughtful question. In this work, we use fixed patch features extracted from a pretrained encoder, as our contribution focuses specifically on the condensation stage rather than on representation learning. This setup also allows a clean comparison across condensation methods by holding the encoder constant.
>
> Because NICER operates on the feature bag produced by the encoder, it can mitigate moderate noise through its pattern - concept mechanism, which naturally filters redundant or unstable patches. However, as with any downstream method, poor feature quality ultimately limits the informativeness of the learned concepts, and thus performance may degrade if the encoder is weak.
>
> We agree that jointly learning the encoder and NICER end-to-end is a promising future direction. Such an approach could further improve feature quality and alignment with condensation. We did not pursue it in this paper because it would introduce a __substantial computational overhead__ - replacing NICER’s lightweight learnable parameters with a large neural encoder - making it difficult to compare fairly with existing condensation baselines. Exploring end-to-end or multi-task extensions is an exciting avenue for future work.

---

> ### Author Response · Authors · 2025-11-24
>
> __Q5. Position & Context. “Position & Context: NICER currently treats the WSI as an unordered bag of patch features, focusing on morphological content. Might incorporating spatial context improve the prototypes (for example, ensuring that “concepts” correspond to contiguous regions or specific structures in tissue)? Some recent works add coordinate information or model the WSI as a graph to capture architecture. Did you consider extensions of NICER to encode spatial relationships between patches or to enforce that selected prototypes are spatially diverse? This could be relevant for tasks like tumor localization, so we wonder if it’s a plausible future direction.”__
>
> We appreciate the reviewer’s insightful question. NICER is intentionally designed to condense patches based solely on morphological similarity, regardless of where they appear in the slide. Because patches exhibiting the same tissue phenotype can be spatially dispersed, ignoring spatial coordinates is a reasonable and effective choice for the condensation problem we target.
>
> That said, incorporating spatial context is a promising extension. Spatial regularization could encourage nearby patches to share concepts or promote spatial coherence when such structure is biologically meaningful. We believe the benefit would be __task-dependent__, for example, tumor localization or architectural phenotype discovery may profit more from spatial constraints than global slide-level prediction.
>
> As the reviewer notes, graph-based or coordinate-aware models would require __spatial structure in the condensed space__, not just in the raw WSI. Designing a spatially structured condensation mechanism that preserves locality while still achieving strong compression is an exciting future direction, and we appreciate the suggestion.
>
> __Q6. Comparison with other Unsupervised Methods You included an “Infinite GPFA” baseline (Yu et al., 2025) which, like NICER, aims to learn latent factors without fixing their number. That method underperformed significantly. Can you shed light on why NICER achieves better results than InfiniteGPFA or other nonparametric clustering approaches? For example, is it due to NICER’s alternating optimization capturing more variance, or the specific way NICER handles patch-to-prototype assignments (top-κ redundancy, etc.)? A deeper explanation would highlight what design choices are most critical for NICER’s success relative to earlier approaches."__
>
> We thank the reviewer for highlighting this point. As pointed out in our answer for Q2, NICER outperforms InfiniteGPFA and other nonparametric clustering methods primarily due to (1) its separation of pattern exploration and concept condensation, and (2) its alternating optimization scheme.
>
> __First__, InfiniteGPFA and similar nonparametric Bayesian models apply a single-level factorization directly on all patch features. This often leads to over-clustering or noisy latent factors because every patch contributes equally to the global model. In contrast, NICER introduces a __pattern exploration stage__ that selects a small set of informative pattern vectors using top-$\kappa$ retrieval. This preserves meaningful morphological variation while filtering out noise and redundant patches before condensation, giving NICER a much cleaner set of inputs for concept condensation.
>
> __Second__, NICER’s __alternating optimization__ lets the model iteratively adjust both patterns and concepts in response to each other. If pattern selection is imperfect or condensation is too coarse, NICER can revisit the original features and refine assignments. InfiniteGPFA’s posterior-based updates do not allow this direct feedback loop, so early clustering errors tend to propagate, often leading to suboptimal or overly fragmented latent factors.
>
> Together, these design choices - structured redundancy in pattern exploration and iterative refinement through alternating optimization - enable NICER to capture more discriminative WSI structure than existing nonparametric clustering or factor models.

---

> ### Author Response · Authors · 2025-11-24
>
> __Q7. Clinical alignment of prototypes. "Given that histologic subtyping is a task pathologists perform visually, have you examined whether the discovered prototypes correspond to interpretable histopathologic structures or features that pathologists recognize? For instance, do the top-activating patches for individual prototypes map to tumor, stroma, or inflammatory regions in a way that aligns with diagnostic reasoning? Such validation would help contextualize the model’s relevance to practical pathology workflows."__
>
> We thank the reviewer for raising this point. Following PANTHER’s evaluation protocol, we visualize the patch-concept assignments and overlay them on the original WSIs (see this link: ​​[link](https://github.com/anon-user-0/NICER-submission/blob/main/analysis_results/Biology_explain/bio.jpg) ). These visualizations show that NICER produces __clearer and more coherent region separation__, indicating that NICER groups morphologically similar patches more faithfully. This aligns with established observations in the WSI literature that similar tissue structures tend to cluster spatially [3].
>
> However, we agree that assigning definitive biological semantics to each prototype requires expert pathological review. While this is beyond the scope of the current paper, we view it as a valuable direction for future collaboration with certified pathologists. For this work, we focus on demonstrating NICER’s __novel formulation, efficiency benefits, and strong empirical performance__, which hold independently of detailed biological annotation.
>
> [3] Ho et al., Deep Learning-Based Objective and Reproducible Osteosarcoma Chemotherapy Response Assessment and Outcome Prediction. Am J Pathol. 2023
>
> __Q8. Cross-institutional validation. "TCGA datasets contain known batch effects and pre-analytic heterogeneity. Have you tested NICER’s generalization when trained on one institution and tested on another (e.g., PANDA KRLS → RUMC split or external private cohorts)? If not, could the authors discuss how NICER’s adaptive, nonparametric mechanism might mitigate or exacerbate domain shifts across institutions?"__
>
> We thank the reviewer for the suggestion. To further assess the generalization of NICER in a cross-cohort setting, we extend our experiments to the CPTAC LUAD survival dataset. Following PANTHER’s preprocessing and evaluation protocol, we train on TCGA-LUAD and test on CPTAC. The resulting C-Index comparisons across representative baselines are reported below.
>
> | Method  | ABMIL        | DSMIL        | ILRA          |
> |---------|--------------|--------------|---------------|
> | KMeans  | 51.51 ± 4.58 | 47.08 ± 2.35 | 55.36 ± 8.19  |
> | OT      | 57.28 ± 2.27 | 49.36 ± 2.41 | 59.00 ± 2.49  |
> | PANTHER | 51.26 ± 2.61 | 46.66 ± 6.29 | 51.69 ± 10.68 |
> | NICER   | 62.36 ± 2.06 | 56.71 ± 1.18 | 62.14 ± 2.96  |
>
> As expected, NICER outperforms other baselines due to its ability to capture distributional information with flexible learning capacity. This aligns with our observation in the experimental results where NICER generalizes well on both condensation and prototyping tasks.
>
> ---
>
> Given the above, we hope the Reviewer can take another look and consider upgrading the rating if our responses have addressed all concerns sufficiently. Otherwise, please let us know if you still have questions for us

---

### Official Review · Reviewer_t1kv · 2025-11-08

**Soundness:** 2
**Presentation:** 3
**Contribution:** 2
**Rating:** 2
**Confidence:** 5

**Summary:**

This work introduces NICER, a data condensation method that compresses bag-of-feature inputs (modeling whole-slide images in pathology) into a compact set of slide-adaptive prototypes for downstream use. NICER works by first learning a high-capacity set of patterns that each patch selects via top-k similarity, then condensing those patterns into a smaller set of concepts while pruning unused concepts which adapts to slide complexity. Benchmark tasks include classification(TCGA-NSCLC, PANDA) and survival prediction (TCGA-LUAD, TCGA-BRCA) with comparisons against unsupervised prototyping baselines (e.g., DeepSets, ProtoCounts, H2T, PANTHER) and MIL predictors (e.g., ABMIL, DSMIL, ILRA). Additional experiments include sensitivity to the initial number of patterns and ablating $k$ in top-$k$ selection.

**Strengths:**

- Good presentation and figures.
- Good number of ablation studies performed (performance trade-off, prototype diversity, $k$ in top-$k$).
- All comparisons use same feature encoder, with ablations of MIL architecture.
- Related work in pathology is well-cited.

**Weaknesses:**

- Study design follows that of PANTHER in evaluating on challenging pathology tasks (PANDAS, survival tasks) with PANTHER being one of the primary comparisons. However, only a few survival tasks are evaluated, with missing evaluation on external datasets such as CPTAC which was one of the core strengths of PANTHER as a prototypical method. Can NICER also generalize to CPTAC for LUAD survival?
- Method is presented nicely but missing many references to data condensation methods. How much of the method comes from existing ideas  in fundamental ML/AI? Very hard to understand the technical contribution.
- From my understanding of this work, NICER does not produce a slide-level representation similar to PANTHER. Rather, it learns a more compact set of patch feature prototypes followed by applying a MIL architecture on top.
- - More fundamental baselines that this work should compare against is k-means, adaptive clustering methods, gaussian / dirchlet process mixture models and other EM clustering ideas in reducing the WSI to a fixed set of prototypes. there exist many non-parameter approaches for solving this same task of pruning redundant clusters in clustering problems.
- - Which formulation of PANTHER Is being compared?

Overall, I don't see the novelty of NICER at this time. Core idea of reducing redundant concepts in unsupervised clustering has a very straightforward extension to pathology, with many fundamental baselines missing that can also be used to find efficient prototype sets. In terms of experimental design, NICER does not evaluate on external datasets for survival prediction, so it is unclear how NICER would generalize.

**Questions:**

See above.

---

> ### Author Response · Authors · 2025-11-24
>
> We thank the reviewer for spending their time on reviewing our work.
>
> __Q1. “Study design follows that of PANTHER in evaluating on challenging pathology tasks (PANDAS, survival tasks) with PANTHER being one of the primary comparisons. However, only a few survival tasks are evaluated, with missing evaluation on external datasets such as CPTAC which was one of the core strengths of PANTHER as a prototypical method. Can NICER also generalize to CPTAC for LUAD survival?”__
>
> We thank the reviewer for the suggestion. To further assess the generalization of NICER in a cross-cohort setting, we extend our experiments to the CPTAC LUAD survival dataset. Following PANTHER’s preprocessing and evaluation protocol, we train on TCGA-LUAD and test on CPTAC. The resulting C-Index comparisons across representative baselines are reported below.
>
> | Method  | ABMIL        | DSMIL        | ILRA          |
> |---------|--------------|--------------|---------------|
> | KMeans  | 51.51 ± 4.58 | 47.08 ± 2.35 | 55.36 ± 8.19  |
> | OT      | 57.28 ± 2.27 | 49.36 ± 2.41 | 59.00 ± 2.49  |
> | PANTHER | 51.26 ± 2.61 | 46.66 ± 6.29 | 51.69 ± 10.68 |
> | NICER   | 62.36 ± 2.06 | 56.71 ± 1.18 | 62.14 ± 2.96  |
>
> As expected, NICER outperforms other baselines due to its ability to capture distributional information with flexible learning capacity. This aligns with our observation in the experimental results where NICER generalizes well on both condensation and prototyping tasks.
>
>
> __Q2. “Method is presented nicely but missing many references to data condensation methods. How much of the method comes from existing ideas in fundamental ML/AI? Very hard to understand the technical contribution.”__
>
> We would like to expand our literature review on existing data condensation as follows. To begin with, we note that data distillation involves crafting a compact yet effective surrogate dataset for a large-scale original database, improving training efficiency. The surrogate is optimized to retain essential information from the cumbersome original, enabling models trained on its to achieve comparable to those trained on the complete one.
>
> __Coreset Selection__[1,2]. This method involves selecting a coreset of the original dataset that ideally contains the entire representativeness of the population. However, this approach encounters a significant performance drop when compression ratio, due to nature heterogeneity in natural data instances.
>
> __Synthesis Direct Optimization__[3]. Wang et al. is the first to propose data condensation, a method that directly optimizes synthetic data instances by iterative optimization between a fixed neural network and the learnable data instances. Due to the limited guidance, this approach often leads to suboptimal performance.
>
> __Gradient-matching__[4,5,6]. This method is one of the main research directions in data condensation, which  improves the supervision by aligning the models’ gradients. Its success has inspired further research into matching the trajectory of gradients[7,8] based on class-specific distinctive information.
>
> __Challenges__. Although existing solutions in data condensation algorithms have advanced the research community, they have two primary limitations: (1) requiring supervision signal to condense large-scale training data to a synthetic set of prototypes, typically encoding class-wise information; and (2) assuming a fixed number of synthetic data instances for the given dataset. These limitations prevent existing solutions from being adapted to WSI settings, where each WSI itself is a dataset - bag of feature instances - with substantial cross-slide heterogeneity.
>
> __Our Contributions__. We propose a novel framework that (1) condenses each WSI into a concept prototype set with no supervision signal and (2) infer an adaptive number of concepts across WSIs based on their complexity. As shown in Table 1, NICER can perform the data condensation task without a supervision signal. For this reason, our technical contributions are novel and valuable even in the ML/AI community.
>
> [1] Bachem et al., “Practical coreset constructions for machine learning”. arXiv.
>
> [2] Sener et al., “Active learning for convolutional neural networks: A coreset approach”. ICLR 2018.
>
> [3] Wang et al., “Dataset Distillation”, arXiv 2018
>
> [4] Shin et al., “Loss-curvature matching for dataset selection and condensation”. AISTATS 2023
>
> [5] Wang et al., “Cafe: Learning to condense dataset by aligning features”, CVPR 2022
>
> [6] Lee et al., “Dataset condensation with contrastive signals”. ICML 2022
>
> [7] Du et al., “Minimizing the accumulated trajectory error to improve dataset distillation”. CVPR 2023
>
> [8] Du et al., “Sequential subset matching for dataset distillation”. NeurIPS 2023.

---

> ### Author Response · Authors · 2025-11-24
>
> __Q3. “From my understanding of this work, NICER does not produce a slide-level representation similar to PANTHER. Rather, it learns a more compact set of patch feature prototypes followed by applying a MIL architecture on top.”.__
>
> We thank the reviewer for the comment. We believe there may be a small misunderstanding. __NICER does produce a slide-level representation__: after condensation, each slide is represented by its resulting sequence of concepts, which serves as the compact slide-level feature set.
>
> This is analogous to PANTHER, which forms slide-level features by concatenating its mixture components in a GMM framework. The difference is that NICER’s representation is nonparametric and adaptive - the number of concepts adjusts to slide complexity - while still producing a valid slide-level input that can be directly consumed by standard WSI MIL decoders. However, we will revise our final version for more clarity.
>
> __Q4.”More fundamental baselines that this work should compare against is k-means, adaptive clustering methods, gaussian / dirchlet process mixture models and other EM clustering ideas in reducing the WSI to a fixed set of prototypes. there exist many non-parameter approaches for solving this same task of pruning redundant clusters in clustering problems.”__
>
> Although our primary goal and inference framework differ from clustering-based approaches, we include comparisons with clustering algorithms as suggested by the reviewer - specifically K-Means, Dirichlet Process K-Means[1] (DPMeans), and Dirichlet Process Mixtures[2] (DPM) - for comprehension.
>
> | Method  | Decoder | Kappa         | Bal. Acc.     | F1            |
> |---------|---------|---------------|---------------|---------------|
> | KMeans  | ABMIL   | 84.57 ± 1.94  | 92.52 ± 1.53  | 92.59 ± 1.51  |
> | DPMeans |         | 83.98 ± 4.39  | 92.00 ± 2.09  | 91.96 ± 2.25  |
> | DPM     |         | 24.03 ± 4.18  | 62.00 ± 2.16  | 61.59 ± 2.51  |
> | NICER   |         | 89.73 ± 1.81  | 94.84 ± 0.92  | 94.87 ± 0.91  |
> | KMeans  | DSMIL   | 81.42 ± 1.10  | 90.73 ± 0.54  | 90.70 ± 0.56  |
> | DPMeans |         | 81.43 ± 1.12  | 90.78 ± 0.60  | 90.70 ± 0.56  |
> | DPM     |         | 16.26 ± 6.20  | 58.14 ± 3.06  | 56.96 ± 4.44  |
> | NICER   |         | 84.61 ± 1.56  | 92.30 ± 0.74  | 91.34 ± 0.79  |
> | KMeans  | ILRA    | 52.96 ± 16.00 | 76.64 ± 7.88  | 74.74 ± 9.56  |
> | DPMeans |         | 40.51 ± 27.49 | 70.44 ± 13.81 | 65.37 ± 19.55 |
> | DPM     |         | 8.64 ± 2.24   | 54.30 ± 1.08  | 54.00 ± 1.76  |
> | NICER   |         | 88.47 ± 1.56  | 94.28 ± 0.76  | 94.23 ± 0.79  |
>
> As expected, these methods capture only coarse representative features and fail to retain the information most relevant for training and generalization. Consequently, they perform significantly worse than NICER and seem to be biased by different decoders. This further emphasizes NICER’s superiority for WSI condensation and its effectiveness when used in downstream predictive tasks.
>
> [1] Kulis et al., Revisiting k-means: New Algorithms via Bayesian Nonparametrics, ICML 2012
>
> [2] Blei, D. M., & Jordan, M. I. (2006). Variational inference for Dirichlet process mixtures.
>
> __Q5. Formulation of PANTHER for comparison__
>
> We compare NICER to PANTHER using the best-performing configuration reported by its authors, namely PANTHER$_{All}$ as described in the original paper. This is the full formulation that uses all prototype components and corresponds to PANTHER’s main objective.
>
> ---
>
> Given the above, we hope the Reviewer can take another look and consider upgrading the rating if our responses have addressed all concerns sufficiently. Otherwise, please let us know if you still have questions for us

---

### Note · Authors · 2025-12-07

I have read and agree with the venue's withdrawal policy on behalf of myself and my co-authors.